# REFINING CLIP'S SPATIAL AWARENESS: A VISUAL-CENTRIC PERSPECTIVE

**Congpei Qiu**[1] **Yanhao Wu**[1] **Wei Ke**[1] **Xiuxiu Bai**[1] **Tong Zhang**[2,3‡]
[1]School of Software Engineering, Xi'an Jiaotong University, China
[2]School of Computer and Communication Sciences, EPFL, Switzerland
[3]University of Chinese Academy of Sciences, Beijing, China

## ABSTRACT

Contrastive Language-Image Pre-training (CLIP) excels in global alignment with language but exhibits limited sensitivity to spatial information, leading to strong performance in zero-shot classification tasks but underperformance in tasks requiring precise spatial understanding. Recent approaches have introduced Region-Language Alignment (RLA) to enhance CLIP's performance in dense multimodal tasks by aligning regional visual representations with corresponding text inputs. However, we find that CLIP ViTs fine-tuned with RLA suffer from notable loss in spatial awareness, which is crucial for dense prediction tasks. To address this, we propose the **Spatial Correlation Distillation (SCD)** framework, which preserves CLIP's inherent spatial structure and mitigates above degradation. To further enhance spatial correlations, we introduce a lightweight **Refiner** that extracts refined correlations directly from CLIP before feeding them into SCD, based on an intriguing finding that CLIP naturally capture high-quality dense features. Together, these components form a robust distillation framework that enables CLIP ViTs to integrate both visual-language and visual-centric improvements, achieving state-of-the-art results across various open-vocabulary dense prediction benchmarks.[1]

## 1 INTRODUCTION

CLIP models (Radford et al., 2021; Sun et al., 2023) have significantly advanced vision-language alignment, achieving notable zero-shot classification and cross-modal retrieval performance. These models align image-level representations with text embeddings, enabling descriptions of wider categories through language. This capability has driven the development of Open-Vocabulary (OV) dense prediction, which aims to recognize a broad range of visual concepts beyond predefined categories. Recent works (Liang et al., 2023; Xu et al., 2023a;b) have successfully extended CLIP's zero-shot abilities to OV dense prediction tasks using Vision Transformer (ViT) models (Dosovitskiy et al., 2021). However, CLIP's image-level pre-training limits its spatial precision in dense cross-modal tasks (Minderer et al., 2022; Paiss et al., 2023). To address this, several approaches (Mukhoti et al., 2023; Zhong et al., 2022; Wu et al., 2023c;b) enhance CLIP's fine-grained cross-modal perception by aligning region-level visual representations with language supervision, a technique known as Region-Language Alignment (RLA), extending CLIP's success to dense prediction tasks.

While acknowledging prior successes, we step back from RLA's focus on language alignment to critically re-examine it from a visual-centric perspective by removing supervision from text. In dense prediction tasks, learning features with strong spatial awareness[2] for localization and recognition is essential (Caron et al., 2021; Oquab et al., 2023; Wu et al., 2023d). Since OV dense prediction tasks extend their visual counterparts, we argue that spatial awareness in CLIP's image encoder is equally crucial. In Fig. 1(a), we analyze the spatial structure of CLIP's dense features using t-SNE (Van der Maaten & Hinton, 2008), and apply unsupervised segmentation with CAUSE (Kim et al., 2023d) as a quantitative measure. Our preliminary findings indicate that RLA strategies, such as RegionCLIP (Zhong et al., 2022) and CLIPSelf (Wu et al., 2023b), result in a notable degradation

---

[1]Code will be available at https://congpeiqiu.github.io/Refining    ‡ Corresponding author.
[2]It refers to the understanding of the spatial relationships between visual concepts within an image.

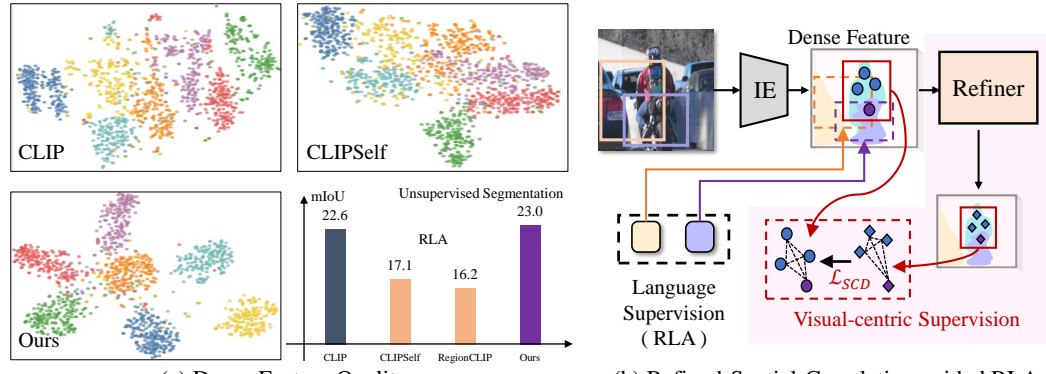

(a) Dense Feature Quality        (b) Refined-Spatial-Correlation guided RLA

Figure 1: **(a) Evaluation of dense feature quality.** We visualize the object-level dense features of image encoder with t-SNE and present the unsupervised segmentation results. Existing Region-Language Alignment methods lead to significant degradation of visual-centric feature quality. **(b) The framework of our fine-tuning structure.** We design an additional visual-centric branch for RLA to enhance model's spatial awareness.

in the visual-centric quality of dense features. We attribute it to the lack of spatial granularity in language supervision, which compromises the model's ability to rich visual-centric perception, rendering RLA methods suboptimal for OV dense prediction tasks. Given these insights, our objective is to improve models spatial awareness during the RLA process, enhancing OV dense prediction from both visual-centric and vision-language perspectives.

In this paper, we propose a Spatial-Correlation-guided Region-Language Alignment (**SC-RLA**) framework, designed to preserve the spatial awareness of CLIP ViTs during the RLA process. One key challenge is domain conflict, as the RLA process projects dense visual embeddings into a text-oriented domain, making them incompatible with visual-centric objectives. To address this, we extend the correlation distillation mechanism (Li et al., 2020; Zhang & Ma, 2023), which focuses on preserving the consistency of spatial relationships between visual concepts encoded by the dense features, to the cross-modal domain, enabling the transfer of visual-centric spatial knowledge. Specifically, we distill spatial correlations from the original CLIP ViT into the student model, enforcing consistency in spatial correlations during fine-tuning and thereby preserving the model's spatial awareness.

While our experiments validate the effectiveness of SC-RLA in preserving CLIP's spatial awareness, a significant limitation persists: CLIP's native spatial awareness remains suboptimal (Wei et al., 2023), which consequently constrains the full potential of SC-RLA. To mitigate this issue, we propose a self-supervised refinement mechanism aimed at enhancing the spatial awareness of CLIP ViTs, thereby improving the supervision quality of SC-RLA. This approach is motivated by a key observation: CLIP ViTs exhibit strong inherent spatial awareness if irrelevant semantic contaminants of CLIP's feature map are filtered out. Building on this insight, we introduce a lightweight module, the ***Refiner***, which generates high-quality spatial refinements from the frozen CLIP ViTs. This process unlocks the dense perception capabilities of the model in a visual-centric manner, without requiring external supervision. By integrating the Refiner into the SC-RLA pipeline, we present R-SC-RLA, a robust framework that enhances CLIP ViTs from both visual-centric and vision-language perspectives.

The effectiveness of our method is experimentally validated on the open-vocabulary dense prediction tasks, including object detection and image segmentation. With only a few epochs of finetuning on small datasets like COCO (Lin et al., 2014), our method achieves non-trivial performance improvements when integrated with the recent RLA methods like CLIPSelf (Wu et al., 2023b) and RegionCLIP (Zhong et al., 2022) for object detection tasks. For the segmentation benchmarks, our method also improves the performance of the recent state-of-the-art model Cat-Seg (Cho et al., 2023).

## 2 RELATED WORK

**Open-vocabulary Dense Prediction.** A rich body of research has focused on refining and transferring the knowledge learned by CLIP (Radford et al., 2021) to downstream tasks. Our approach targets

two key areas within open-vocabulary dense prediction: object detection and image segmentation. In object detection, two primary strategies are commonly used: i) designing additional network structures for object localization while utilizing the Vision-Language Model (VLM) encoder as a feature extractor for region-language alignment (Wu et al., 2023c; Minderer et al., 2022; Kuo et al., 2022), and ii) extending conventional detection models by learning from VLM-provided region-language alignment signals through distillation (Du et al., 2022; Ma et al., 2022; Wang et al., 2023; Pham et al., 2024; Wu et al., 2023a; Gu et al., 2021). Segmentation, which requires finer-grained cross-modal alignment, has advanced in parallel with object detection. Similar to detection strategies, segmentation can be addressed by generating class-agnostic masks while leveraging VLM's vision-to-text matching capabilities (Xu et al., 2023a; 2022; Yu et al., 2024; Ding et al., 2022), or by distilling cross-modal consistency knowledge into existing segmentation models (Chen et al., 2023a;b; Qin et al., 2023). Despite the success of these methods, they remain tailored to specific tasks. To enable broader applications, our approach focuses on fine-tuning the CLIP image encoder at the midstream stage to improve generalizability.

**Region-Language alignment.** Inspired by the success of language-image alignment (Radford et al., 2021; Kim et al., 2021; Li et al., 2022a), considerable attention has been directed toward facilitating RLA at various training stages. At the upstream pre-training stage, some studies introduce region-text alignment tasks using annotated visual grounding data (Li et al., 2022b; Liu et al., 2023), or generate pseudo-region-level text annotations from image captions (Zhong et al., 2022). At the midstream stage, to avoid large-scale pre-training from scratch, several works (Mukhoti et al., 2023; Wu et al., 2024a; Zhou et al., 2022a; Lin et al., 2023) refine image-level vision-language correspondence into a form more suitable for dense-level tasks. This is achieved by training a lightweight RLA module (Mukhoti et al., 2023), extracting training-free RLA signals (Zhou et al., 2022a), or fine-tuning the image encoder (Lin et al., 2023; Wu et al., 2024a). The recent advance of CLIPSelf (Wu et al., 2023b) enhances RLA by directly aligning region representations with the text-oriented $[CLS]$ token of the image encoder, eliminating the need for text. Since recent OV dense prediction models combines dense prediction with vision-text matching, improving the spatial awareness of the image encoder is as critical as enhancing its alignment with language signals—an aspect seldom discussed in previous RLA research and a key motivation for our work.

**Correlation Distillation.** Correlation distillation(Gao et al., 2022a; Li et al., 2020; Zhang & Ma, 2023; Peng et al., 2019; 2023; Yang et al., 2022) is commonly utilized to ensure consistency of structural correlations within feature representations between target and source feature sets. This approach typically employs a correlation matrix, either within the same feature map (Peng et al., 2023; Yang et al., 2022) or across different instances (Gao et al., 2022a; Li et al., 2020; Peng et al., 2019; Zhang & Ma, 2023), to capture these structural dependencies, which are then used to supervise the distillation process. In our work, we harness the spatial awareness of CLIP by leveraging spatial correlation to guide Region-Language Alignment. We demonstrate the feasibility and robustness of correlation as an effective tool for bridging the cross-modal gap, enabling vision-language models to benefit from a visual-centric perspective. Unlike conventional methods (Peng et al., 2023; Li et al., 2020; Peng et al., 2019; Zhang & Ma, 2023), our approach is unique in its multi-modal focus, utilizing spatial correlation to improve open-vocabulary dense prediction tasks.

## 3 METHODOLOGY

### 3.1 PRELIMINARY: REGION-LANGUAGE ALIGNMENT

**Region-Language Alignment.** Let CLIP's image encoder be denoted as $f_I$, with an input image $\boldsymbol{X}$ and a set of region proposals $\{\boldsymbol{b}_i\}_{i=1}^{B}$. Region-Language Alignment (RLA) methods fine-tune the student model $f_I^s$, initialized from $f_I$, to align region representations with corresponding language supervision. This alignment is achieved using the following loss function:

$$\mathcal{L}_{\text{RLA}} = \frac{1}{B} \sum_i \mathcal{L}_{\text{Align}}(\text{RoIPooling}(f_I^s(\boldsymbol{X}), \boldsymbol{b}_i), \boldsymbol{T}_i), \tag{1}$$

where $\boldsymbol{T}_i$ denotes the language supervision corresponding to region $\boldsymbol{b}_i$, and $\mathcal{L}_{\text{Align}}$ represents an alignment loss, such as InfoNCE (Oord et al., 2018) or cosine similarity. As depicted in the top left of Fig. 2, we explore two key RLA mechanisms from RegionCLIP (Zhong et al., 2022) and CLIP-Self (Wu et al., 2023b). RegionCLIP aligns region proposals with object nouns to generate pseudo

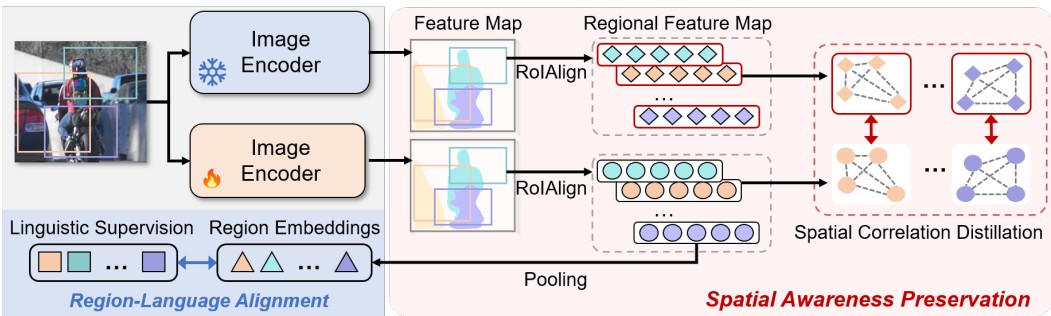

Figure 2: **Overview of SC-RLA.** The conventional RLA process (blue arrow) aligns the region representations of the student model with the corresponding language supervision signals generated by either CLIP's text encoder or image encoder. We enhance this process by integrating Spatial Correlation Distillation (red arrow) to preserve the structural relationships between visual tokens.

region-text pairs, which are processed by the text encoder to obtain $\boldsymbol{T}_i$. We adapt RegionCLIP's RLA process for fine-tuning following the approach of (Wu et al., 2023b), which we term 'RegionText'. In contrast, CLIPSelf leverages the inherent consistency between image encoder's $[CLS]$ tokens and text embeddings, using the $[CLS]$ tokens of cropped images defined by $\boldsymbol{b}_i$ as the corresponding $\boldsymbol{T}_i$.

**Limitation of RLA.** As shown in Fig. 1(a), RLA compromises the visual-centric quality of dense features for the alignment with the language domain (full results and technical details are provided in Appendix A). However, we argue that OV dense prediction requires a dual capability: strong consistency with language, and robust **spatial awareness** for dense prediction. Prioritizing only one dimension, as RLA does, is suboptimal. To address this, we propose a visual-centric solution that seamlessly integrates with RLA to effectively balance both aspects.

### 3.2 SPATIAL-CORRELATION-GUIDED RLA

To enhance spatial awareness, one might consider integrating dense-level visual pre-training techniques (Wang et al., 2021; Zhou et al., 2022b) or aligning the dense features of the student and teacher models. However, these approaches conflict with RLA's goal, which projects visual-centric dense features into the language domain. To reconcile this, we introduce **S**patial **C**orrelation **D**istillation (**SCD**), inspired by correlation distillation methods (Li et al., 2020; Peng et al., 2019), as shown in the bottom right of Fig. 2. To capture region-level semantics, we process the input image $\boldsymbol{X}$ through both the student model $f_I^s$ and teacher model $f_I$, extracting regional features $\boldsymbol{Z}_i^s, \boldsymbol{Z}_i^t \in \mathbb{R}^{L \times D}$ with sampled proposals $\{\boldsymbol{b}_i\}_{i=1}^{B}$ using RoIAlign (He et al., 2017), where $L$ denotes the sequence length of the flattened dense features. This process is formulated as:

$$\boldsymbol{Z}_i^s = \text{RoIAlign}(f_I^s(\boldsymbol{X}), \boldsymbol{b}_i), \ \boldsymbol{Z}_i^t = \text{RoIAlign}(f_I(\boldsymbol{X}), \boldsymbol{b}_i). \tag{2}$$

The spatial correlation matrices $\boldsymbol{C}_i^s, \boldsymbol{C}_i^t \in \mathbb{R}^{L \times L}$ are then computed as:

$$\boldsymbol{C}_i^s = \boldsymbol{Z}_i^s \cdot (\boldsymbol{Z}_i^s)^T, \ \boldsymbol{C}_i^t = \boldsymbol{Z}_i^t \cdot (\boldsymbol{Z}_i^t)^T. \tag{3}$$

We normalize these matrices using softmax to highlight regional structural relationships:

$$\hat{\boldsymbol{C}}_i^s(j, k; \tau_s) = \frac{\exp(\boldsymbol{C}_i^s(j, k)/\tau_s)}{\sum_{k'} \exp(\boldsymbol{C}_i^s(j, k')/\tau_s)}, \ \hat{\boldsymbol{C}}_i^t(j, k; \tau_t) = \frac{\exp(\boldsymbol{C}_i^t(j, k)/\tau_t)}{\sum_{k'} \exp(\boldsymbol{C}_i^t(j, k')/\tau_t)}, \tag{4}$$

where $\tau$ is a temperature parameter, and $\boldsymbol{C}_i(j, k)$ is the element at coordinate $(j, k)$. To preserve spatial awareness of the student model, we minimize the cross-entropy loss between the student and teacher correlation matrices:

$$\mathcal{L}_{\text{SCD}} = \frac{1}{B} \sum_i \frac{1}{L} \sum_j H(\hat{\boldsymbol{C}}_i^s(j, :), \hat{\boldsymbol{C}}_i^t(j, :)). \tag{5}$$

Since $\mathcal{L}_{\text{SCD}}$ focuses solely on spatial correlations without requiring cross-domain consistency, it integrates smoothly with RLA, guiding the fine-tuning process from a visual-centric perspective. This leads to the **SC-RLA** objective:

$$\mathcal{L}_{\text{SC-RLA}} = \mathcal{L}_{\text{RLA}} + \lambda \mathcal{L}_{\text{SCD}}, \tag{6}$$

where $\lambda$ is a hyperparameter that balances the two losses.

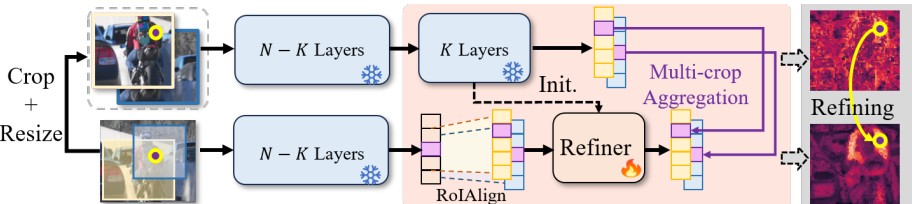

Figure 3: **A training-free illustration of refining CLIP.** We compute the average features from a frozen CLIP model across diverse contexts to mitigate semantic contamination. As the number of aggregated images $N$ increases, the model's spatial awareness improves progressively.

Figure 4: **CLIP refining pipeline.** The proposed pipeline enhances CLIP's dense representations using a lightweight Refiner module. Initialized with the last $K$ layers of CLIP's image encoder, this module aggregates corresponding tokens in a global-to-local dynamic, eliminating unnecessary contextual distortion and focusing on high-quality local semantics.

## 3.3 REFINING SPATIAL AWARENESS OF CLIP

As demonstrated in Sec.4, the SC-RLA objective significantly improves the OV dense prediction. However, CLIP's inherent spatial awareness remains limited(Wei et al., 2023). To further enhance the SCD process, we propose to explicitly refine CLIP's spatial awareness.

**Identifying CLIP's Dense-level Potential.** Our approach is driven by a key observation: CLIP inherently provides robust dense representations for vision-centric perception tasks. To substantiate this, we conduct a training-free investigation, as illustrated in Fig. 3. Given a set of randomly sampled images $\{\boldsymbol{X}_i\}_{i=1}^N$, we embed a predefined target image $\boldsymbol{X}_t$ into each $\boldsymbol{X}_i$ at random positions, producing modified images $\boldsymbol{X}_i^M$ with $\boldsymbol{X}_i$ serving as the context. These modified images are processed through CLIP to extract the submap $\boldsymbol{Z}_{\boldsymbol{X}_t|\boldsymbol{X}_i}$ corresponding to $\boldsymbol{X}_t$. We then refine the target's features by averaging the submaps, yielding an aggregated feature map $\bar{\boldsymbol{Z}}_{\boldsymbol{X}_t}$:

$$\bar{\boldsymbol{Z}}_{\boldsymbol{X}_t} = \frac{1}{N} \sum_i \boldsymbol{Z}_{\boldsymbol{X}_t|\boldsymbol{X}_i}. \tag{7}$$

In this setup, the target image $\boldsymbol{X}_t$ remains constant across all $\boldsymbol{X}_i^M$, with the only variation being the context provided by the different $\boldsymbol{X}_i$. Compared to the direct output from CLIP, the aggregated feature map $\bar{\boldsymbol{Z}}_{\boldsymbol{X}_t}$, especially for larger $N$, is more focused on fine-grained semantics. This finding reveals a critical insight: CLIP's dense features are subject to semantic contamination from contextual information. By aggregating features from different contexts, we can effectively mitigate these distortions. Further analysis, detailed in Appendix B, demonstrates that the refined features significantly enhance performance in dense prediction tasks.

**Refining CLIP's Dense-level Representation.** The analysis indicates that enhancing CLIP's spatial awareness in a visual-centric manner is achievable. However, aggregating large numbers of images is computationally expensive and impractical for inference. Therefore, to explicitly extract high-quality dense features at once, we propose to train a lightweight *Refiner* module. It leverages the insight of the above analysis, but performs aggregation within the same image, as depicted in Fig. 4. For the frozen CLIP image encoder $f_I := f_I^B \circ f_I^A$, where $f_I^B$ ($f_I^A$) represent the final $K$(initial $N - K$) residual blocks of $f_I$, we initialize the *Refiner* $f_R$ by cloning $f_I^B$. Given an input image $\boldsymbol{X}$ and a selected region $\boldsymbol{b}$, $f_R$ outputs the refined feature map as:

$$\hat{\boldsymbol{Z}} = f_R \left( \text{RoIAlign}(f_I^A(\boldsymbol{X}), \boldsymbol{b}) \right). \tag{8}$$

Here, $f_R$ inherits the knowledge learned by $f_I^B$ and is fine-tuned to extract spatially aware refinements from the output of the frozen $f_I^A$. To train the Refiner, we diverge from the common local-to-global

Table 1: **Zero-shot evaluation of dense representation.** We report Top1 and Top5 mean accuracy.

| Backbone | Method | RPN Proposals | Boxes | | Thing Masks | | Stuff Masks | |
|---|---|---|---|---|---|---|---|---|
| | | | Top1 | Top5 | Top1 | Top5 | Top1 | Top5 |
| ViT-B/16 | EVA-CLIP | - | 18.2 | 33.2 | 20.6 | 36.5 | 18.4 | 43.5 |
| ViT-B/16 | CLIPSelf | ✗ | 72.1 | 91.3 | 74.4 | 91.8 | 46.8 | 80.2 |
| ViT-B/16 | R-SC-CLIPSelf | ✗ | 76.0 | 93.1 | 76.2 | 92.5 | **53.5** | **84.4** |
| ViT-B/16 | RegionText | ✓ | 71.1 | 90.7 | 73.7 | 91.4 | 34.2 | 68.6 |
| ViT-B/16 | R-SC-RegionText | ✓ | 72.0 | 91.3 | 74.3 | 91.6 | 41.6 | 73.3 |
| ViT-B/16 | CLIPSelf | ✓ | 74.0 | 92.6 | 76.3 | 92.8 | 36.8 | 75.0 |
| ViT-B/16 | R-SC-CLIPSelf | ✓ | **77.3** | **94.0** | **78.9** | **94.2** | 52.6 | 83.9 |
| ViT-L/14 | EVA-CLIP | - | 56.7 | 78.0 | 59.0 | 79.8 | 20.8 | 41.9 |
| ViT-L/14 | CLIPSelf | ✗ | 77.1 | 93.3 | 78.7 | 93.7 | 44.4 | 78.3 |
| ViT-L/14 | R-SC-CLIPSelf | ✗ | 82.9 | 96.0 | 82.8 | 95.6 | **57.8** | **86.5** |
| ViT-L/14 | CLIPSelf | ✓ | 77.8 | 94.0 | 80.4 | 94.5 | 34.0 | 71.8 |
| ViT-L/14 | R-SC-CLIPSelf | ✓ | 81.7 | 95.8 | **82.9** | **95.9** | 52.5 | 83.9 |

approach in self-supervised learning (Zhang et al., 2022; Caron et al., 2021) and instead design a global-to-local alignment mechanism. This eliminates unnecessary contextual distortion outside a local region, enabling the network to focus on high-quality, fine-grained semantics, similar to the aggregation process in Eq. 7. Specifically, we randomly sample local region proposals $\{b'_i\}_{i=1}^C$ to generate $C$ local crops $X'_i$ from $X$. We then forward the global image $X$ and the region $b'_i$ through Eq. 8 to obtain refinements $\hat{Z}_i \in \mathbb{R}^{L \times D}$, and pass the context-free local crops $X'_i$ through $f_I$ to extract local feature maps $Z'_i \in \mathbb{R}^{L \times D}$. We align the corresponding tokens between $\hat{Z}i$ and $Zi'$, defining the Refining loss as:

$$\mathcal{L}_{\text{Refiner}} = \frac{1}{C} \sum_i \mathcal{L}_{align}(\hat{Z}_i, Z'_i), \tag{9}$$

where $\mathcal{L}_{align}$ denotes the alignment loss. In our implementation, we use InfoNCE (Oord et al., 2018) for $\mathcal{L}_{align}$ due to its robustness, treating other tokens within the same crop as negative samples. A detailed analysis of the alignment loss is provided in Appendix C.2.

### 3.4 REFINED SPATIAL CORRELATION DISTILLATION

**Overall Framework.** To enhance CLIP's spatial awareness using the trained Refiner, we modify the target correlation matrix in Eq. 3 by replacing $Z^t_i$ with the refined features $\hat{Z}_i$. This allows us to supervise the spatial correlations in the student model using the refined spatial structure. We refer to this process as **R**efined **S**patial **C**orrelation **D**istillation (**R-SCD**), which forms the final R-SC-RLA framework. Notably, the refined model does not participate in the RLA branch, thereby preserving the integrity of the language supervision.

**Visual-centric Application.** The R-SCD process can also be applied independently to the student model, focusing solely on enhancing spatial awareness without language supervision. We call this approach **V**isual-centric R-SCD (**R-SC-V**).

## 4 EXPERIMENTAL RESULTS

### 4.1 IMPLEMENTATION DETAILS

Our full distillation consists of two stages: i) the refining of *Refiner*; and ii) CLIP fine-tuning stage. Although the two stages can be jointly trained in an end-to-end manner (Sec. 4.5), we first train i) to obtain a stable Refiner, then utlize the refinements to guide ii). Concretely, we use 8 RTX 3090 GPUs for both stages with AdamW (Loshchilov & Hutter, 2017) optimizer. For the first stage, we set the learning rate to $1e-4$ and train Refiner for 4 epochs with the batch size as 16. For the second stage, we set the learning rate to $2e-5$ and perform CLIP fine-tuning for 6 epochs with the batch size as 4. The proposals for RLA process are generated by a trained RPN, identical to (Wu et al., 2023b). Both stage are trained on COCO *train2017* dataset (Lin et al., 2014). The experiments involves two

Table 2: **Effects of Refiner.** Comparison of distilled models with and without refining.

| Backbone | Method | RPN Proposals | Boxes | | Thing Masks | | Stuff Masks | |
|---|---|---|---|---|---|---|---|---|
| | | | Top1 | Top5 | Top1 | Top5 | Top1 | Top5 |
| ViT-B/16 | CLIPSelf | ✓ | 74.0 | 92.6 | 76.3 | 92.8 | 36.8 | 75.0 |
| ViT-B/16 | SC-CLIPSelf | ✓ | 76.0 | 93.5 | 77.9 | 93.9 | 49.4 | 82.6 |
| ViT-B/16 | R-SC-CLIPSelf | ✓ | **77.3** | **94.0** | **78.9** | **94.2** | **52.5** | **83.9** |

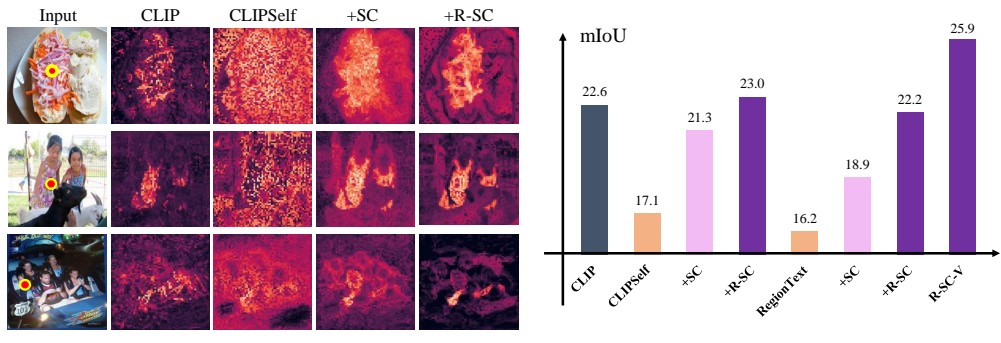

(a) Point-affinity Visualization | (b) Unsupervised Segmentation

Figure 5: **Visual-centric analysis.** (a) We visualize the affinity map $w.r.t$ a selected query token embeddings (marked by the red dot) of the visual encoder. (b) Unsupervised segmentation evaluation with CAUSE on Cityscapes, where the mIoU is reported.

CLIP models: OpenAI CLIP (Radford et al., 2021) and EVA-CLIP (Sun et al., 2023). For our design specifics, Refiner is initialized with the weights of the last 4 blocks of the visual encoder for ViT-B and the last 6 blocks for ViT-L, with the early layers kept frozen. To optimize Refiner, we generate $C = 4$ crops per image at scale ratios between $[0.3, 0.7]$. During the stage of spatial correlation distillation, we set the temperature $\tau_T = \tau_S = 0.2$, with $\lambda = 0.2$ for ViT-B and $\lambda = 0.4$ for ViT-L. Further structural details of Refiner are deferred to the Appendix. C.

## 4.2 EVALUATION OF DENSE REPRESENTATION

**Recognition Capability.** We conduct dense-level zero-shot classification to evaluate model recognition capabilities, following the protocol in (Wu et al., 2023b). The region representations are extracted with three strategies: i) *Boxes*, which applies RoIPooling to COCO dataset object bounding boxes, ii) *Thing Masks*, and iii) *Stuff Masks*, both extracted via mask pooling (He et al., 2017) using COCO Panoptic dataset masks (Kirillov et al., 2019). The results are shown in Tab. 1, where 'RegionText' refers to RegionCLIP's RLA process. Our method yields consistent and significant improvements across all settings. As shown in Tab. 2, we further demonstrate the Refiner's necessity. Notably, R-SC-RLA achieves a $10\% - 20\%$ improvement on COCO-Stuff using RPN proposals, where many objects are neglected by the RLA supervision. This indicates that SCD can still effectively transfer language supervision to tokens, even when they are misaligned with the text.

**Visual-centric Analysis.** From a visual-centric perspective, we access the quality of the dense features both qualitatively and quantitatively to analyze the causes of above improvements. The visualization of point affinity maps is shown in Fig. 5, following the principle in (Bai et al., 2022) (full results provided in Appendix. F), where we calculate the cosine similarity map between a selected token and the feature map. Additionally, we use CAUSE for unsupervised segmentation on Cityscapes (Cordts et al., 2016) as a quantitative indicator. Both results demonstrate a significant improvement regarding to the quality of dense features, which is consistent with our motivation of enhancing model's spatial awareness.

## 4.3 OPEN-VOCABULARY DENSE PREDICTION

We evaluate the fine-tuned models via OV dense predction, including detection on OV-COCO and OV-LVIS benchmarks following the protocol in (Wu et al., 2023b), and semantic segmentation following Cat-Seg (Cho et al., 2023). The corresponding details are presented in the Appendix. D.

Table 3: **Results on open-vocabulary object detection.** We report $AP_{50}^{novel}$ of the novel classes for OV-COCO and $mAP_r$ of the rare classes for OV-LVIS. 'SC-' denotes employing SC-RLA, and 'R-SC-' denotes the full distillation strategy wtih the Refiner.

(a) OV-COCO benchmark

| Method | Backbone | $AP_{50}^{novel}$ |
|---|---|---|
| F-VLM (Kuo et al., 2022) | RN50 | 28.0 |
| BARON-KD (Wu et al., 2023a) | RN50 | 34.0 |
| LP-OVOD (Pham et al., 2024) | RN50 | 40.5 |
| ViLD (Gu et al., 2021) | RN50 | 27.6 |
| Detic (Zhou et al., 2022c) | RN50 | 27.8 |
| RegionCLIP (Zhong et al., 2022) | RN50×4 | 39.3 |
| CORA (Wu et al., 2023c) | RN50×4 | 41.7 |
| CORA+ (Wu et al., 2023c) | RN50×4 | 43.1 |
| PromptOVD (Song & Bang, 2023) | ViT-B/16 | 30.6 |
| RO-ViT (Kim et al., 2023b) | ViT-L/16 | 33.0 |
| CFM-ViT (Kim et al., 2023a) | ViT-L/16 | 34.1 |
| DITO (Kim et al., 2023c) | ViT-L/16 | 46.1 |
| RegionText | ViT-B/16 | 34.4 |
| SC-RegionText | ViT-B/16 | 35.8 |
| R-SC-RegionText | ViT-B/16 | 37.0 |
| CLIPSelf (Wu et al., 2023b) | ViT-B/16 | 37.6 |
| SC-CLIPSelf | ViT-B/16 | 39.1 |
| R-SC-CLIPSelf | ViT-B/16 | 40.9 |
| CLIPSelf (Wu et al., 2023b) | ViT-L/14 | 44.3 |
| SC-CLIPSelf | ViT-L/14 | 46.5 |
| R-SC-CLIPSelf | ViT-L/14 | **48.1** |

(b) OV-LVIS benchmark

| Method | Backbone | $mAP_r$ |
|---|---|---|
| BARON-KD (Wu et al., 2023a) | RN50 | 22.6 |
| OV-DETR (Zang et al., 2022) | RN50 | 17.4 |
| Detic (Zhou et al., 2022c) | RN50 | 24.9 |
| CORA+ (Wu et al., 2023c) | RN50×4 | 28.1 |
| F-VLM (Kuo et al., 2022) | RN50×4 | 32.8 |
| VLDet (Lin et al., 2022) | SwinB | 26.3 |
| Detic (Zhou et al., 2022c) | SwinB | 33.8 |
| PromptOVD (Song & Bang, 2023) | ViT-B/16 | 23.1 |
| RO-ViT (Kim et al., 2023b) | ViT-B/16 | 28.4 |
| RO-ViT (Kim et al., 2023b) | ViT-L/16 | 32.4 |
| CFM-ViT (Kim et al., 2023a) | ViT-B/16 | 28.8 |
| CFM-ViT (Kim et al., 2023a) | ViT-L/16 | 33.9 |
| DITO (Kim et al., 2023c) | ViT-L/16 | **38.4** |
| CoDet (Ma et al., 2023) | ViT-L/14 | 37.0 |
| RegionText | ViT-B/16 | 21.2 |
| R-SC-RegionText | ViT-B/16 | 23.6 |
| CLIPSelf (Wu et al., 2023b) | ViT-B/16 | 25.3 |
| R-SC-CLIPSelf | ViT-B/16 | 27.5 |
| CLIPSelf (Wu et al., 2023b) | ViT-L/14 | 34.9 |
| R-SC-CLIPSelf | ViT-L/14 | 37.2 |

Table 4: **Results on open-vocabulary segmentation.** We report the mIoU performance. † denotes the vanilla version of Cat-Seg.

| Method | VLM | ADE-150 | | ADE-847 | | PASCAL Context | |
|---|---|---|---|---|---|---|---|
| | | mIoU | mACC | mIoU | mACC | mIoU | mACC |
| SAN (Xu et al., 2023b) | CLIP ViT-B/16 | 27.5 | 45.6 | 10.1 | 21.1 | 53.8 | 73.0 |
| SAN (Xu et al., 2023b) | CLIP ViT-L/14 | 32.1 | 50.7 | 12.4 | 25.2 | 57.7 | 77.6 |
| SILC (Naeem et al., 2023) | SILC-C-B/16 | 37.0 | - | 13.5 | - | 61.2 | - |
| SILC (Naeem et al., 2023) | SILC-C-L/16 | 37.7 | - | 15.0 | - | 63.5 | - |
| Cat-Seg† | CLIP ViT-B/16 | 27.2 | 41.2 | 8.4 | 16.6 | 57.5 | 74.0 |
| Cat-Seg†+CLIPSelf | CLIP ViT-B/16 | 29.0 | 46.0 | 9.3 | 20.1 | 58.0 | 75.3 |
| Cat-Seg†+R-SC-CLIPSelf | CLIP ViT-B/16 | 29.9 | 47.2 | 9.8 | 21.2 | 58.3 | 75.9 |
| Cat-Seg | CLIP ViT-B/16 | 31.8 | 48.8 | 12.0 | 22.6 | 57.5 | 75.5 |
| Cat-Seg+CLIPSelf | CLIP ViT-B/16 | 30.8 | 48.4 | 11.9 | 21.9 | 56.3 | 75.0 |
| Cat-Seg+R-SC-CLIPSelf | CLIP ViT-B/16 | 32.0 | 48.9 | 12.2 | 22.0 | 57.2 | 75.3 |
| Cat-Seg+R-SC-V | CLIP ViT-B/16 | 32.7 | 49.7 | 12.3 | 22.6 | 58.0 | 76.0 |
| Cat-Seg | CLIP ViT-L/14 | 37.9 | 55.7 | 16.0 | 28.7 | 63.3 | 80.0 |
| Cat-Seg+R-SC-V | CLIP ViT-L/14 | **38.4** | **56.0** | **16.6** | **29.2** | **63.6** | **80.2** |

**Open-vocabulary Object Detection.** Following (Wu et al., 2023b), we utilize a two-stage detector, *F-ViT*, which extracts multi-scale feature maps from the intermediate layers of the frozen EVA-CLIP model. We report the $AP50^{novel}$ for novel classes on the OV-COCO dataset and $mAP_r$ for rare classes on the OV-LVIS dataset, with results presented in Tab. 3. When combined with a RLA method, such as CLIPSelf or RegionText, our SCD module consistently enhances performance, achieving a final improvement of $2\% - 4\%$ across all benchmarks when further integrated with the Refiner.

**Open-vocabulary Semantic Segmentation.** Cat-Seg, a state-of-the-art model for open-vocabulary semantic segmentation, leverages OpenAI's CLIP ViTs as its vision-language backbone, followed by a cost-aggregation module. We evaluate two variants: the original Cat-Seg with a frozen text encoder, and an updated version with a fine-tuned text encoder. Trained on the ADE20K dataset (Zhou et al., 2017) and evaluated on ADE-847, ADE-150, and Pascal Context (Mottaghi et al., 2014), our distilled model—enhanced with R-SCD objective—consistently outperforms both Cat-Seg and CLIPSelf in the vanilla setup, as shown in Tab. 4. Interestingly, fine-tuning the text encoder in the updated Cat-Seg results in a performance decline for CLIPSelf. We attribute this to the fine-tuned text encoder achieving more precise implicit region-language alignment, thus diminishing CLIPSelf's advantage.

Figure 6: Off-the-shelf segmentation with MaskCLIP.

| VLM Model | PASCAL Context | COCO Stuff |
|---|---|---|
| OpenAI-CLIP | 25.5 | 14.6 |
| +CLIPSelf | 26.4 | 16.1 |
| +R-SC-CLIPSelf | **27.9** | **17.5** |
| DFN | 29.4 | 18.6 |
| +CLIPSelf | 30.8 | 20.1 |
| +R-SC-CLIPSelf | **32.1** | **21.2** |
| Meta-CLIP | 30.3 | 20.0 |
| +CLIPSelf | 30.1 | 19.7 |
| +R-SC-CLIPSelf | **33.6** | **22.0** |
| EVA-CLIP | 22.8 | 15.6 |
| +CLIPSelf | 32.2 | 20.1 |
| +R-SC-CLIPSelf | **37.0** | **23.8** |

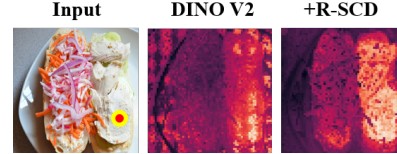

Figure 7: **Visualization of segmentation results.** We visualize the segmentation results with MaskCLIP using different VLM backbones. Best viewed in color and zoomed in.

Table 5: **Unsupervised segmentation with CAUSE.** We report mIoU and mACC results.

| Dataset | Method | mIoU | mACC |
|---|---|---|---|
| Cityscapes | DINO V2 | 29.9 | 89.8 |
| | + R-SC-V | **31.8** | **90.5** |
| COCO-Stuff | DINO V2 | 43.0 | 76.9 |
| | + R-SC-V | **44.1** | **77.4** |

Figure 8: **Affinity map visualization** of the given red point on DINOv2 and DINOv2+R-SCD. Lighter regions indicate higher affinity.

Furthermore, CLIPSelf's limited spatial awareness contributes to this decline. To address these issues, we employ the R-SC-V objective, described in Sec. 3.3, as a visual-centric fine-tuning strategy, which leads to superior performance across all datasets.

**Off-the-shelf Zero-shot Segmentation.** We further apply our method to more CLIP's variants, including DFN (Fang et al., 2024) and Meta-CLIP (Xu et al., 2024). We adopt the off-the-shelf segmentation protocol in MaskCLIP (Zhou et al., 2022a), which directly classifies each dense feature output by the frozen image encoder using cosine similarity with the corresponding category embedded by the text encoder. The mIoU results are reported in Tab. 6, showcasing the superiority and generalizability of our method. Visualization is provided in Fig. 7, with more examples in Fig. 17.

## 4.4 VISUAL-CENTRIC APPLICATION: ENHANCING DINO V2

DINO V2 (Oquab et al., 2023) is a self-supervised foundational model designed for vision-centric tasks. However, as highlighted by (Darcet et al., 2023), DINO V2 tends to produce dense feature artifacts, which impair its ability to capture fine-grained details and result in abnormal representations dominated by global context. To address these shortcomings, we integrate R-SC-V as a visual-centric enhancement module to fine-tune DINO V2. This enhancement consistently improves performance in unsupervised segmentation tasks, as evidenced by results on the Cityscapes (Cordts et al., 2016) and COCO-Stuff (Caesar et al., 2018) datasets (see Tab. 5). Moreover, the failure cases observed in DINO V2, visualized in Fig. 8, are notably reduced after R-SC-V fine-tuning.

## 4.5 ABLATION STUDY

We dissect our framework and study the impact of each component to reveal the strengths of our designs. A more comprehensive investigation can be found in the Appendix. E.

**Comparison with Correlation Distillation.** We compare only the SCD method, excluding the Refiner, against several established techniques in correlation distillation (Li et al., 2020; Peng et al., 2019; 2023; Yang et al., 2022). To adjust the distillation objective, we replace the standard cross-entropy loss with the Frobenius norm of the correlation matrix, following the approach in (Yang et al., 2022; Li et al., 2020), which we denote as $\mathcal{L}_F$. In terms of correlation matrix construction, we explore two alternatives: $\mathcal{L}_{Inter}$, which emphasizes inter-instance correlations across various feature maps (Peng et al., 2019); and $\mathcal{L}_{Attn}$, which focuses on attention values (Peng et al., 2023). Results in

Table 6: **Ablation on the design choices of R-SCD.** We report Top1 for zero-shot dense prediction and $AP_{50}^{novel}$ for OV-COCO.

(a) Ablation on SCD

| Method | Boxes $Top1$ | Stuff $Top1$ | Thing $Top1$ | OV-COCO |
|---|---|---|---|---|
| CLIPSelf | 74.0 | 76.3 | 36.8 | 37.6 |
| *(Correlation distillation designs)* | | | | |
| $+\mathcal{L}_F$ | 73.5 | 75.2 | 35.9 | 36.8 |
| $+\mathcal{L}_{Inter}$ | 73.4 | 75.4 | 37.2 | 37.2 |
| $+\mathcal{L}_{Attn}$ | 74.3 | 76.2 | 36.6 | 37.9 |
| *(Different visual-centric constraints)* | | | | |
| $+\mathcal{L}_{CL}$ | 65.1 | 67.6 | 29.6 | 27.4 |
| $+\mathcal{L}_{MIM}$ | 73.6 | 75.9 | 36.3 | 37.5 |
| $+\mathcal{L}_{SCD}$ | 76.0 | 77.9 | 49.4 | 39.1 |

(b) Ablation on Refiner

| Method | Boxes $Top1$ | Thing $Top1$ | Stuff $Top1$ | OV-COCO |
|---|---|---|---|---|
| CLIPSelf | 74.0 | 76.3 | 36.8 | 37.6 |
| w/ R-SCD | 76.0 | 77.9 | 49.4 | 39.1 |
| *(Designs of Refiner)* | | | | |
| +Random | 76.7 | 78.3 | 50.8 | 39.4 |
| +Exogenous | 76.8 | 78.2 | 51.4 | 39.9 |
| +PACL | 76.4 | 77.6 | 49.9 | 39.2 |
| *(Training strategies)* | | | | |
| w/ R-SCD (E2E) | 76.8 | 78.4 | 51.9 | 40.0 |
| w/ R-SCD (L2G) | 75.2 | 76.9 | 43.2 | 38.0 |
| +Ours | 77.3 | 78.9 | 52.5 | 40.9 |

Tab. 6(a) indicate that our method, which prioritizes structural relationships within the same scene, is more effective at enhancing spatial awareness during RLA fine-tuning.

**Comparison on Visual-centric Constraints.** Previous work has utilized visual-centric self-supervised learning techniques (He et al., 2022; Chen et al., 2020; Zhou et al., 2022b) to improve the dense feature quality of CLIP (Dong et al., 2023; Li et al., 2023). However, these methods are limited to image-language pre-training, where fine-grained language supervision is not a concern. This raises the question of whether they are suitable for RLA fine-tuning, as discussed in Sec. 3.2. Following MaskCLIP(Dong et al., 2023), we incorporate an additional EMA model, updated via momentum from the student's weights to provide visual supervision. We explore two types of constraints: (i) $\mathcal{L}_{MIM}$, which adopt masked image modeling objective as iBOT (Zhou et al., 2022b); and (ii) $\mathcal{L}_{CL}$, with dense-level contrastive loss in DenseCL (Wang et al., 2021). As shown in Tab. 6(a), these constraints fail to improve the performance, supporting our claim that typical visual-centric constraints may conflict with dense-level language supervision without non-trivial modifications.

**Ablation on the Refiner's Structure.** We evaluate different architectural designs for the Refiner: (i) *Random Initialization*, where no weights are inherited from the final $K$ attention blocks of CLIP; (ii) *Exogenous*, where a randomly initialized Refiner is applied on top of CLIP; and (iii) *PACL*, which integrates a lightweight residual block (vision embedder) as proposed in PACL (Mukhoti et al., 2023). Table Tab. 6(b) demonstrates that all Refiner variants improve performance, highlighting the importance of the refinement process. Our approach, which leverages the weights from the last $K$ attention blocks of CLIP, achieves the best results, underscoring the benefit of inheriting pretrained knowledge from CLIP for the Refiner module.

**Global-to-Local Refining Dynamics.** We further investigate the impact of the global-to-local dynamics on training the Refiner. As illustrated in Table 6(b), a local-to-global (L2G) pipeline reversing the process in Fig. 4 leads to significant performance degradation, compared to SC-CLIPSelf without the Refiner. This confirms the necessity of our global-to-local design.

**End-to-end Training.** In Table 6(b), we present results from end-to-end (E2E) training, where both the Refiner and the student encoder are fine-tuned simultaneously. Although the performance is slightly lower than that of the two-stage training approach, it still surpasses the CLIPSelf and SC-CLIPSelf baselines, demonstrating the flexibility of our framework. Nevertheless, we recommend the two-stage training method in practice for optimal performance.

## 5 CONCLUSION

In this paper, we introduced the Spatial Correlation Distillation framework to address the issue of quality degradation in dense features when fine-tuning CLIP ViTs with Region-Language Alignment. Our approach preserves the spatial structural knowledge of the model and incorporates the Refiner module to further enhance CLIP's spatial awareness, leading to notable performance gains on open-vocabulary dense prediction benchmarks. Our work highlights the critical role of spatial awareness in vision-language models from a visual-centric perspective, extending beyond mere linguistic alignment. The experimental results demonstrate that our framework enables CLIP ViTs to integrate both vision-language and visual-centric enhancements, providing a novel avenue for advancing dense-level perception in CLIP-based models.

ACKNOWLEDGMENTS

This work was supported in part by the National Natural Science Foundation of China under Grant No. 62376209 and 62472349.

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

APPENDIX CONTENTS

The appendix is structured as follows:

- Appendix A presents comprehensive experiments evaluating the dense features of various fine-tuned CLIP ViTs.
- Appendix B provides an analysis of the dense features from the original CLIP ViTs, serving as empirical evidence for our refining strategy.
- Appendix C details the design of the Refiner, supplemented with ablation studies and further empirical study.
- Appendix D outlines the implementation details for the open-vocabulary dense prediction tasks.
- Appendix E includes additional ablation studies for the overall framework.
- Appendix F provides the implementation details for point-affinity visualization, along with further visual results.

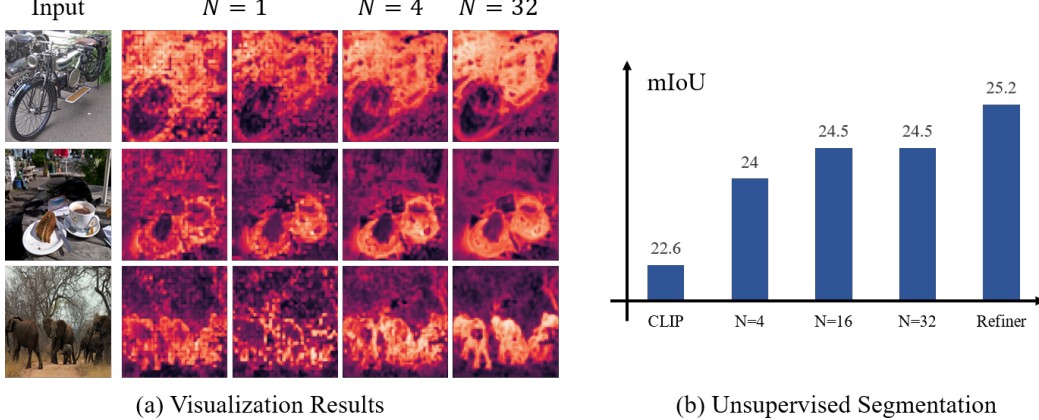

(a) Visualization Results        (b) Unsupervised Segmentation

Figure 9: **(a)** point-affinity visualization with different number $N$ of aggregated images. An increasing $N$ tends to rendering dense features with better spatial awareness. Best viewed in color and zoomed in. **(b)** When the semantic contamination of dense features is effectively eliminated with a large $N$, unsupervised segmentation present significant performance improvement. 'Refiner' denotes utilzing the output of our trained Refiner for inference.

## A  VISUAL-CENTRIC EVALUATION OF DENSE FEATURES

### A.1  UNSUPERVISED SEGMENTATION.

As Oquab et al. (2023) argue, a powerful pre-trained visual encoder can produce dense features that are directly applicable to unsupervised segmentation, even surpassing the performance of fine-tuned methods. Building on this insight, we perform unsupervised segmentation using the state-of-the-art CAUSE (Kim et al., 2023d) as a numerical indicator to assess the quality of the dense features generated by a frozen visual encoder.

### A.2  T-SNE OF DENSE FEATURES.

t-SNE (Van der Maaten & Hinton, 2008) is a widely used technique for projecting high-dimensional embeddings into a lower-dimensional space for visualization. In our experiments, we first extract instance-level features by applying masked average pooling to the dense features generated by an image encoder, using the ground-truth segmentation masks to define the pooling regions. We then apply t-SNE to project the extracted object-level dense features into a 2D space for visualization. To enhance clarity, we randomly sample 256 instances from each category and select 7 categories for each visualization. The images and corresponding annotations are taken from the COCO *train2017* dataset. More visualizations are provided in Fig. 15.

## B  ANALYSIS ON DENSE-LEVEL POTENTIAL OF CLIP

### B.1  EXTRACTING HIGH-QUALITY DENSE FEATURES FROM FROZEN CLIP

As an experimental complement, we present more visualization results in Fig. 9(a), which presents a clearer trend that when the number $N$ of modified images $\boldsymbol{X}^M$ in Fig. 3 increases, the dense features tend to be more spatially aware and aligned with the object boundaries. For the qualitative evaluation, a large $N$ as 32 yields 2% mIoU improvement in the unsupervised segmentation without any training process. This observation demonstrates the dense-level potential of the CLIP image encoder once we eliminate the irrelevant distractions hindering dense feature quality, the aggregation operation acts as an average filter to filter out the semantic contamination.

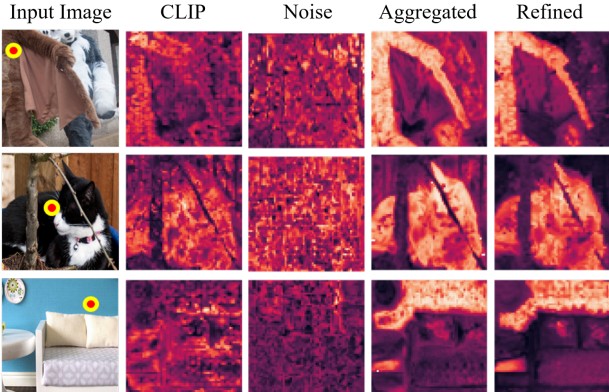

Figure 10: **Point-affinity visualization of dense features.** From left to right: CLIP's original feature map, semantic contamination, aggregated dense features, and the output of the trained Refiner.

### B.2 EFFECTS OF REFINER

If we consider the aggregated features $\bar{Z}_{X_t}$ in Eq. 7 as the target features, for each feature map $Z$ directly output by CLIP image encoder, we define the noise pattern $\epsilon := Z - \bar{Z}_{X_t}$ as the deviation from the target features. As in Fig. 10, the noise results in meaningless correlation, irrelevant to the fine-grained visual concepts. For the effects of our desinged Refiner, Shown in Fig. 9(b), the trained Refiner exhibits more effectiveness in unsupervised segmentation than both the original dense features and the aggregated feature, demonstrating the necessity of Refiner.

## C REFINER

### C.1 DESIGN CHOICES

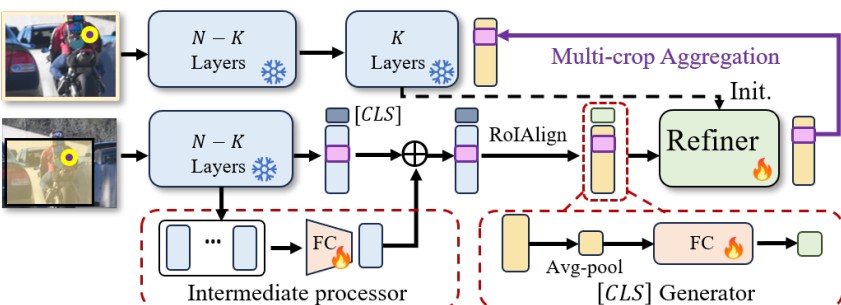

Figure 11: **The architecture of the proposed Refiner.** The framework consists of three components: a Refiner head, an Intermediate processer, and a region-level $[CLS]$ token generator.

The Refiner consists of three components: a Refiner head, an intermediate processor, and a region-level $[CLS]$ token generator. We here detail the design of the latter two components.

**Region-level $[CLS]$ generator.** The $[CLS]$ token of ViT contains the global information of the image input. As the $[CLS]$ is directly bonded with the full image, to integrate with region-level features, the patch tokens in earlier layer output corresponding to region bounding box $b_i$ are fused with RoI pooling and subsequently forwarded to a two-layer MLP with a hidden size of 4096, which is derived as:

$$\hat{z}_i^{[CLS]} = \text{FC}_{\text{CLS}} \left[ \text{RoIPool}(f_I^A(X), b_i) \right]. \tag{10}$$

**Intermediate processer.** To extract refined dense representations from earlier layers $f_A$, instead of solely processing its final outputs, we also utilize the output tokens from the $l_1, l_2$-th layer as the

Table 8: **OV-COCO detection results with different loss.** We report the $\text{AP}_{50}^{novel}$ and $\text{AP}_{50}^{base}$ results.

| Model | Method | OV-COCO | |
| | | $\text{AP}_{50}^{novel}$ | $\text{AP}_{50}^{base}$ |
|---|---|---|---|
| ViT-B/16 | CLIPSelf | 37.6 | 54.9 |
| ViT-B/16 | SCD-Cos | 34.5 | 50.1 |
| ViT-B/16 | SCD-NCE | 40.9 | 54.7 |

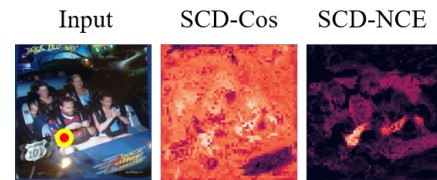

Figure 12: **Affinity map with different loss.** We present the affinity map obtained with Refiner for Cosine loss and InfoNCE loss.

intermediate auxiliary input, $i.e.$:

$$\hat{\boldsymbol{Z}}_i = f_R\left(\text{RoIAlign}(f_I^A(\boldsymbol{X}) + \boldsymbol{Z}_{Inter}, \boldsymbol{b}_i)\right), \boldsymbol{Z}_{Inter} = \text{FC}_{Inter}\left[\text{Concat}\left(\boldsymbol{Z}_{l_1}, \boldsymbol{Z}_{l_2}\right)\right], \quad (11)$$

where the multi-scale processer $\text{FC}_{\text{Inter}} : \mathbb{R}^{2D} \to \mathbb{R}^D$ is a two-layer MLP with a hidden size of 4096. For the visual encoder of ViT-B, we set $l_1 = 4$ and $l_2 = 7$, and for ViT-L, we set $l_1 = 9$ and $l_2 = 14$.

### C.2 MORE ABLATION ON REFINER

**Designs of Refiner.** We dissect the components of the Refiner to investigate their contributions and present the results in Tab. 7. Both the Intermediate processer and the $[CLS]$ generator contribute to the extraction of high-quality refined spatial correlation, which is crucial for the distillation process, thus yielding performance improvement with both components enabled. Additionally, instead of local regions defined by the proposals, we also explore the 'Late' setting where we perform RoIAlign on the output of Refiner, $i.e.$:

Table 7: **Ablation on different components in Refiner.** We report $\text{AP}_{50}^{novel}$ on OV-COCO.

| $\text{FC}_{\text{Inter}}$ | $\text{FC}_{\text{CLS}}$ | Late | $\text{AP}_{50}^{novel}$ |
|---|---|---|---|
| | | | 40.1 |
| $\checkmark$ | | | 40.3 |
| $\checkmark$ | $\checkmark$ | | 40.9 |
| $\checkmark$ | $\checkmark$ | $\checkmark$ | 40.2 |

$$\hat{\boldsymbol{Z}}_i = \text{RoIAlign}\left(f_R(f_I^A(\boldsymbol{X})), \boldsymbol{b}_i\right). \quad (12)$$

However, this setting leads to performance degradation, indicating the necessity to focus model's attention on the local regions.

**Cosine vs. InfoNCE.** Our original Refiner objective with the InfoNCE loss is derived as:

$$\mathcal{L}_{\text{NCE}} = \frac{1}{C'}\sum_i -\frac{1}{L}\sum_{j=1}^{L}\log\frac{\exp(\hat{\boldsymbol{Z}}_i[j]\cdot\boldsymbol{Z}_i'[j])}{\sum_k\exp(\hat{\boldsymbol{Z}}_i[j]\cdot\boldsymbol{Z}_i'[k])}, \quad (13)$$

To demonstrate the necessity of InfoNCE for training the Refiner, we conduct an additional experiment by replacing Eq. 9 with the cosine loss:

$$\mathcal{L}_{\text{Cos}} = \frac{1}{C'}\sum_i -\frac{1}{L}\sum_{j=1}^{L}\cos(\hat{\boldsymbol{Z}}_i[j], \boldsymbol{Z}_i'[j]). \quad (14)$$

We visualize the affinity map calculated with the dense features output by the Refiner in Fig. 12, where the selected token can be entangled with its irrelevant surroundings. This phenomenon harms the Refiner for extracting high-quality refinements, leading to performance drop as presented in Tab. 8. In contrast, the intra-feature-map contrast in Eq. 13 further filters out interference from irrelevant neighboring tokens, effectively tackling this issue.

### C.3 SEMANTIC COUPLING IN CLIP

To further assess whether the Refiner's effects align with its design objectives, we conduct a quantitative analysis to evaluate the tendency of CLIP's dense features to become entangled with irrelevant context, referred to as ***semantic coupling*** as in Fig. 13, which is also observed by recent works (Qiu et al., 2023a;b; Wu et al., 2024b). Specifically, we concatenate two independently sampled images $\boldsymbol{X}_A$ and $\boldsymbol{X}_B$ side by side, denoted as $\boldsymbol{X}_{AB}$, which introduces context disturbance

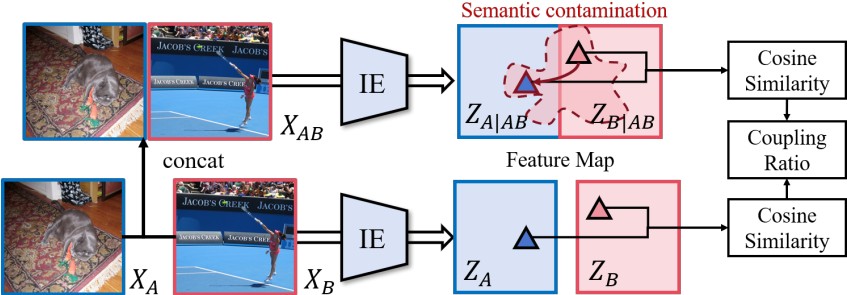

Figure 13: **Measuring pipeline of semantic coupling.** We concatenate two independently sampled images $X_A$ and $X_B$ to analyze the semantic contamination between them. The defined coupling ratio reflects the significance of semantic coupling.

from $X_B$ to $X_A$. We forward $X_{AB}, X_A, X_B$ to the image encoder to obtain regional feature map $Z_{A|AB}, Z_{B|AB}, Z_A, Z_B$. Finally, the coupling ratio is computed as:

$$\text{CR} = \mathbb{E}_i \left[ \frac{cos(Z_{A|AB}[i], Z_{B|AB}[j])}{cos(Z_A[i], Z_B[j])} \right], j = \arg\max_k cos(Z_{A|AB}[i], Z_{B|AB}[k]), \quad (15)$$

where we identify the most similar token $j$ in $Z_{B|AB}$ to the token $i$ in $Z_{A|AB}$, and analyze whether this similarity arises from the entanglement of irrelevant semantics introduced by the concatenation operation. Ideally, the CR value is expected to be close to 1, as $X_A$ and $X_B$ possess independent semantics. By calculating the average CR value across COCO *val2017*, we report the measured CR value in Tab. 9. The results indicate that both the original and CLIPSelf-finetuned CLIP models are significantly affected by semantic coupling. In contrast, our proposed Refiner effectively addresses this issue, demonstrating high consistency with its intended design goals of eliminating semantic contamination.

Table 9: **CR value of different models.** We report the CR values with different finetuning strategies using EVA-CLIP.

| Method | EVA-CLIP | w/ CLIPSelf | EVA-CLIP-Refiner | w/ R-SC-CLIPSelf |
|---|---|---|---|---|
| CR↓ | 2.32 | 1.86 | 0.95 | 0.97 |

# D    IMPLEMENTATION DETAILS OF OPEN-VOCABULARY DENSE PREDICTION

## D.1    OPEN-VOCABULARY OBJECT DETECTION.

We adopt F-ViT (Wu et al., 2023b) as the open-vocabulary object detector, which replaces the simple Feature Pyramid Network (FPN) of ViTDet (Li et al., 2022c) detector with a standard FPN and utilizes the feature maps from multiple intermediate layers of the ViT. The entire visual encoder is keep frozen during the training process. The F-ViT model is trained for 3 epochs for the OV-COCO benchmark and 48 epochs for the OV-LVIS benchmark. Following the common practice, the box AP with IoU threshold of 0.5 on the novel classes is reported for OV-COCO, and the mean mask AP is reported for OV-LVIS.

## D.2    OPEN-VOCABULARY SEMANTIC SEGMENTATION.

We utilize two version of Cat-Seg (Cho et al., 2023) for the open-vocabulary semantic segmentation task. Both the vanilla and updated versions of Cat-Seg fine-tune the attention weights of the vision encoder and the additional cost aggregation module. The main difference at the level of VLM is that the vanilla version freezes the text encoder of CLIP, while the updated version fine-tunes the text encoder to implicitly align the vision and text representations. The model is trained on the ADE20K (Zhou et al., 2017) dataset. We evaluate the model on three benchmarks: A-150 and A-847, which contain 150 and 847 classes respectively, and Pascal Context (Mottaghi et al., 2014) dataset

Table 10: Full comparison on OV-COCO benchmark.

| Method | Backbone | $AP_{50}^{novel}$ | $AP_{50}^{base}$ | $AP_{50}$ |
|---|---|---|---|---|
| OV-RCNN (Zareian et al., 2021) | RN50 | 17.5 | 41.0 | 34.9 |
| RegionCLIP (Zhong et al., 2022) | RN50 | 26.8 | 54.8 | 47.5 |
| RegionCLIP (Zhong et al., 2022) | Rn50 | 31.4 | 57.1 | 50.4 |
| RegionCLIP (Zhong et al., 2022) | RN50x4 | 39.3 | 61.6 | 55.7 |
| ViLD (Gu et al., 2021) | RN50 | 27.6 | 59.5 | 51.2 |
| OV-DETR (Zang et al., 2022) | RN50 | 29.4 | 61.0 | 52.7 |
| PB-OVD (Gao et al., 2022b) | RN50 | 30.8 | 46.1 | 42.1 |
| Detic (Zhou et al., 2022c) | RN50 | 27.8 | 51.1 | 45.0 |
| OC-OVD (Bangalath et al., 2022) | RN50 | 36.6 | 54.0 | 49.4 |
| VLDet (Lin et al., 2022) | RN50 | 32.0 | 50.6 | 45.8 |
| F-VLM (Kuo et al., 2022) | RN50 | 28.0 | - | 39.6 |
| BARON-Cap (Wu et al., 2023a) | RN50 | 33.1 | 54.8 | 49.1 |
| BARON-KD (Wu et al., 2023a) | RN50 | 34.0 | 60.4 | 53.5 |
| BARON-Cap&KD (Wu et al., 2023a) | RN50 | 42.7 | 54.9 | 51.7 |
| OADP (Wang et al., 2023) | RN50 | 35.6 | 55.8 | 50.5 |
| CORA (Wu et al., 2023c) | RN50 | 35.1 | 35.5 | 35.4 |
| CORA (Wu et al., 2023c) | RN50x4 | 41.7 | 44.5 | 43.8 |
| CORA+ (Wu et al., 2023c) | RN50x4 | 43.1 | 60.9 | 56.2 |
| RO-ViT (Kim et al., 2023b) | ViT-B/16 | 30.2 | - | 41.5 |
| RO-ViT (Kim et al., 2023b) | ViT-L/16 | 33.0 | - | 47.7 |
| CFM-ViT (Kim et al., 2023a) | ViT-L/16 | 34.1 | - | 46.0 |
| DITO (Kim et al., 2023c) | ViT-L/16 | 40.8 | - | 50.3 |
| CLIPSelf (Wu et al., 2023b) | ViT-B/16 | 37.6 | 54.9 | 50.4 |
| R-SC-CLIPSelf | ViT-B/16 | 40.9 | 54.7 | 51.1 |
| CLIPSelf (Wu et al., 2023b) | ViT-L/14 | 44.3 | 64.1 | 59.0 |
| R-SC-CLIPSelf | ViT-L/14 | **48.1** | 65.4 | 60.8 |

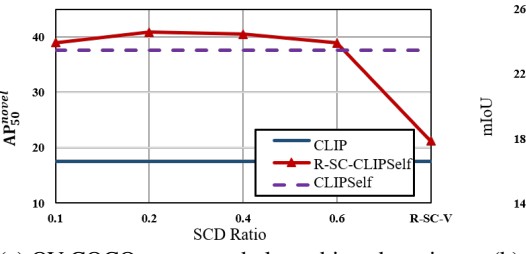 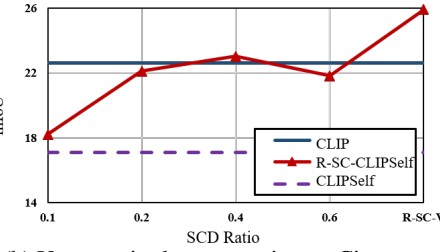

(a) OV-COCO open-vocabulary object detection     (b) Unsupervised segmentation on Cityscapes

Figure 14: **Ablation on spatial correlation distillation.** We control the loss ratio of SCD and report $AP_{50}^{novel}$ on OV-COCO detection and mIoU on Cityscapes segmentation.

with the PC-59 benchmark. The baseline of Cat-Seg is conducted by rerun the training process with the official released code.

## E    FURTHER ABLATION STUDIES

**SCD Ratio $\lambda$.** We conduct an ablation study with various SCD ratios $\lambda$ to investigate the effects of the spatial correlation distillation. We evaluate the performance of the distilled model on two levels: i) the open-vocabulary object detection on OV-COCO and ii) the unsupervised segmentation on Cityscapes (Cordts et al., 2016) with CAUSE (Kim et al., 2023d). All the models are fine-tuned on COCO *train2017* dataset for 6 epochs following the setting of CLIPSelf with proposals, except for 'R-SC-V' that focuses on the visual-centric fine-tuning. As the OVOD task weights more on the vision-to-text alignment capability, we additionally involve the unsupervised segmentation task to evaluate the quality of the dense representations. As depicted in Fig. 14(b), the alignment between

Table 11: Full comparison on OV-LVIS benchmark.

| Method | Backbone | mAP$_r$ | mAP$_c$ | mAP$_f$ | mAP |
|---|---|---|---|---|---|
| RegionCLIP (Zhong et al., 2022) | RN50 | 17.1 | 27.4 | 34.0 | 28.2 |
| RegionCLIP (Zhong et al., 2022) | RN50x4 | 22.0 | 32.1 | 36.9 | 32.3 |
| Detic (Zhou et al., 2022c) | RN50 | 24.9 | - | - | 32.4 |
| Detic (Zhou et al., 2022c) | SwinB | 33.8 | - | - | 47.0 |
| VLDet (Lin et al., 2022) | RN50 | 21.7 | 29.8 | 34.3 | 30.1 |
| VLDet (Lin et al., 2022) | SwinB | 26.3 | 39.4 | 41.9 | 38.1 |
| ViLD (Gu et al., 2021) | RN50 | 16.6 | 24.6 | 30.3 | 25.5 |
| OV-DETR (Zang et al., 2022) | RN50 | 17.4 | 25.0 | 32.5 | 26.6 |
| DetPro (Du et al., 2022) | RN50 | 19.8 | 25.6 | 28.9 | 25.9 |
| BARON-KD (Wu et al., 2023a) | RN50 | 22.6 | 27.6 | 29.8 | 27.6 |
| OADP (Wang et al., 2023) | RN50 | 21.7 | 26.3 | 29.0 | 26.6 |
| OC-OVD (Bangalath et al., 2022) | RN50 | 21.1 | 25.0 | 29.1 | 25.9 |
| F-VLM (Kuo et al., 2022) | RN50 | 18.6 | - | - | 24.2 |
| F-VLM (Kuo et al., 2022) | RN50x4 | 26.3 | - | - | 28.5 |
| F-VLM (Kuo et al., 2022) | RN50x16 | 30.4 | - | - | 32.1 |
| F-VLM (Kuo et al., 2022) | RN50x64 | 32.8 | - | - | 34.9 |
| CORA (Wu et al., 2023c) | RN50x4 | 22.2 | - | - | - |
| CORA+ (Wu et al., 2023c) | RN50x4 | 28.1 | - | - | - |
| OWL-ViT (Kim et al., 2023b) | ViT-L/14 | 25.6 | - | - | 34.7 |
| RO-ViT (Kim et al., 2023b) | ViT-B/16 | 28.0 | - | - | 30.2 |
| RO-ViT (Kim et al., 2023b) | ViT-L/16 | 32.1 | - | - | 34.0 |
| RO-ViT (Kim et al., 2023b) | ViT-H/16 | 34.1 | - | - | 35.1 |
| CFM-ViT (Kim et al., 2023a) | ViT-L/16 | 33.9 | - | - | 36.6 |
| DITO (Kim et al., 2023c) | ViT-L/16 | **38.4** | - | - | 37.7 |
| CoDet (Ma et al., 2023) | ViT-L/14 | 37.0 | - | - | - |
| CLIPSelf (Wu et al., 2023b) | ViT-B/16 | 25.3 | 21.8 | 29.1 | 25.2 |
| R-SC-CLIPSelf | ViT-B/16 | 27.5 | 22.7 | 29.8 | 26.3 |
| CLIPSelf (Wu et al., 2023b) | ViT-L/14 | 34.9 | 34.6 | 35.6 | 35.1 |
| R-SC-CLIPSelf | ViT-L/14 | 37.2 | 37.2 | 37.1 | 37.2 |

the visual and $[CLS]$ token presented by CLIPSelf causes the degradation of the segmentation performance. With the SCD loss that extracts and maintains the spatial correlation, the performance degradation is mitigated, achieving the balance between the vision-to-text alignment and the dense-level understanding. Moreover, when applying the R-SC-V loss, the performance is further improved with a non-trivial margin. In addition, as observed in Fig. 14(a), SCD loss significantly boosts the performance on OV-COCO, even for the R-SC-V model without a RLA branch, indicating the importance of the refined spatial awareness holds for the OVOD task.

**Depth of the Refiner.** We investigate the impact of the depth of the Refiner on the performance of the distilled model. The depth will affect the distillation process from two aspects: i) the balance between the capacity of refining and preserving the original visual knowledge learned by the visual encoder, and ii) the computational efficiency of the training process. We conduct experiments with different depths of the Refiner. A deeper Refiner will increase the parameter size and the complexity of the fine-tuned model, but more difficult to perserve learned knowledge of the pre-trained model. As shown in Tab. 12, the model with a 4-layer Refiner achieves the best performance, obtaining balance between the refining capacity and the knowledge preservation.

**Temperature of Spatial Correlation Distillation.** We conduct an ablation study on the temperature of the spatial correlation distillation as shown in Tab. 13. The temperature $\tau_s$ and $\tau_t$ of the student and teacher logits respectively control the softness of the spatial correlation distillation. Generally, a sharpening process with $\tau_s > \tau_t$ typically leads to higher performance than the distillation with $\tau_s < \tau_t$. But we find an equal temperature setting of $\tau_s = \tau_t = 0.2$ achieves the best performance, which indicates that the denoised spatial correlation stems from the intra-scene contrast loss is already sharp enough.

Table 12: **Ablation on the depth of the Refiner.** We report the $AP_{50}^{novel}$ on OV-COCO and the Top1 performance of zero-shot classification on COCO

| Depth $K$ | OV-COCO $AP_{50}^{novel}$ | Boxes Top1 Acc. | Thing Masks Top1 Acc. | Stuff Masks Top1 Acc. |
|---|---|---|---|---|
| 2 | 40.3 | 76.3 | 78.0 | 50.7 |
| 3 | 40.7 | 77.0 | 78.5 | 52.5 |
| 4 | 40.9 | 77.3 | 78.9 | 52.5 |
| 5 | 40.5 | 76.9 | 78.1 | 51.6 |

Table 13: **Ablation on the temperature of the spatial correlation distillation.** We report the $AP_{50}^{novel}$ on OV-COCO

| $\tau_s$ | $\tau_t$ | OV-COCO $AP_{50}^{novel}$ | $AP_{50}^{base}$ | $AP_{50}$ |
|---|---|---|---|---|
| 0.1 | 0.1 | 39.2 | 54.0 | 50.2 |
| 0.15 | 0.15 | 39.6 | 53.5 | 49.9 |
| 0.2 | 0.15 | 39.2 | 53.3 | 49.6 |
| 0.2 | 0.2 | 40.9 | 54.7 | 51.1 |
| 0.2 | 0.3 | 38.5 | 54.5 | 50.4 |
| 0.25 | 0.25 | 39.8 | 53.9 | 50.2 |

**Local vs. Global Distillation.** For spatial correlation distillation, we utilize $B$ sampled bounding box to define the region for distillation. Here we investigate another setting that directly distills the spatial correlation of the entire image to the student model, *i.e.* defining the region bounding box as the whole image area. The results are presented in Tab. 14. Though still effective with performance improvement, global distillation significantly underperforms local distillation, which aligns with our intuition to facilitate the model to focus on the local.

Table 14: **Comparison of local and global distillation strategy.** We report the $AP_{50}^{novel}$ on OV-COCO and the Top1 performance of zero-shot classification on COCO

| Strategy | OV-COCO $AP_{50}^{novel}$ | Boxes Top1 Acc. | Thing Masks Top1 Acc. | Stuff Masks Top1 Acc. |
|---|---|---|---|---|
| CLIPSelf | 37.6 | 74.0 | 76.3 | 36.8 |
| Local | 40.9 | 77.3 | 78.9 | 52.5 |
| Global | 39.7 | 75.6 | 77.8 | 50.2 |

**Training on larger-scale dataset.** We further fine-tune the Refiner and train EVA-CLIP with R-SC-CLIPSelf on CC3M (Sharma et al., 2018) for one epoch, evaluating the performance using MaskCLIP. As presented in Tab. 15, our model benefits from the larger-scale dataset, achieving improved multi-modal dense prediction performance.

# F  VISUALIZATION

## F.1  AFFINITY MAP

As presented in Fig. 16, the query token is marked with a red dot, and the cosine similarity between the query token and the feature map is calculated for the visualization. We visualize the vanilla CLIP, CLIPSelf, R-SC-CLIPSelf, RegionText, and R-SC-RegionText respectively.

## F.2  MASKCLIP SEGMENTATION

As presented in Fig. 17, we adopt off-the-shelf zero-shot segmentation with MaskCLIP (Zhou et al., 2022a) and present the results of visualization with EVA-CLIP and Meta-CLIP backbones.

Table 15: Off-the-shelf segmentation with MaskCLIP.

| Method | Dataset | PASCAL Context | COCO Stuff |
|---|---|---|---|
| R-SC-CLIPSelf | COCO | 37.0 | 23.8 |
| R-SC-CLIPSelf | CC3M | 38.2 | 25.0 |

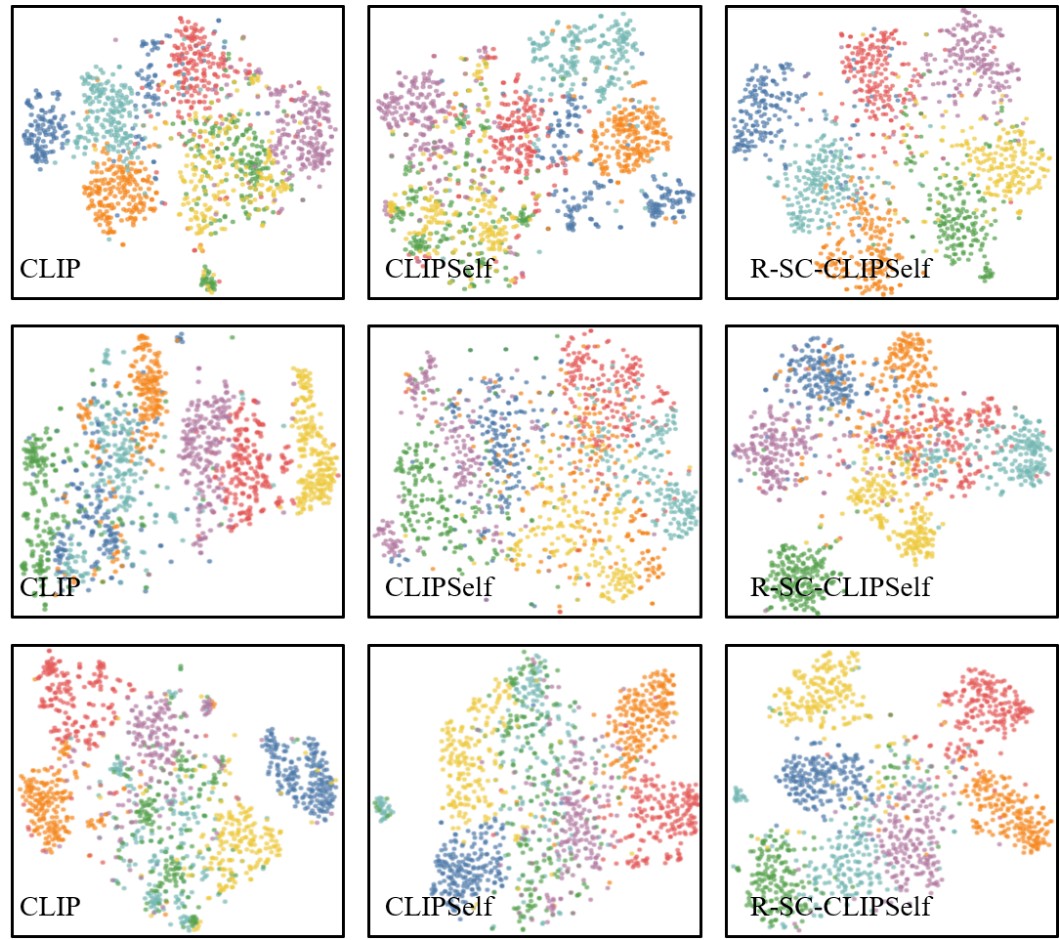

Figure 15: **Visualization of t-SNE.** In each row, we visualize the dense features with the same set of categories. We respectively present the results of vanilla CLIP, CLIPSelf, and R-SC-CLIPSelf.

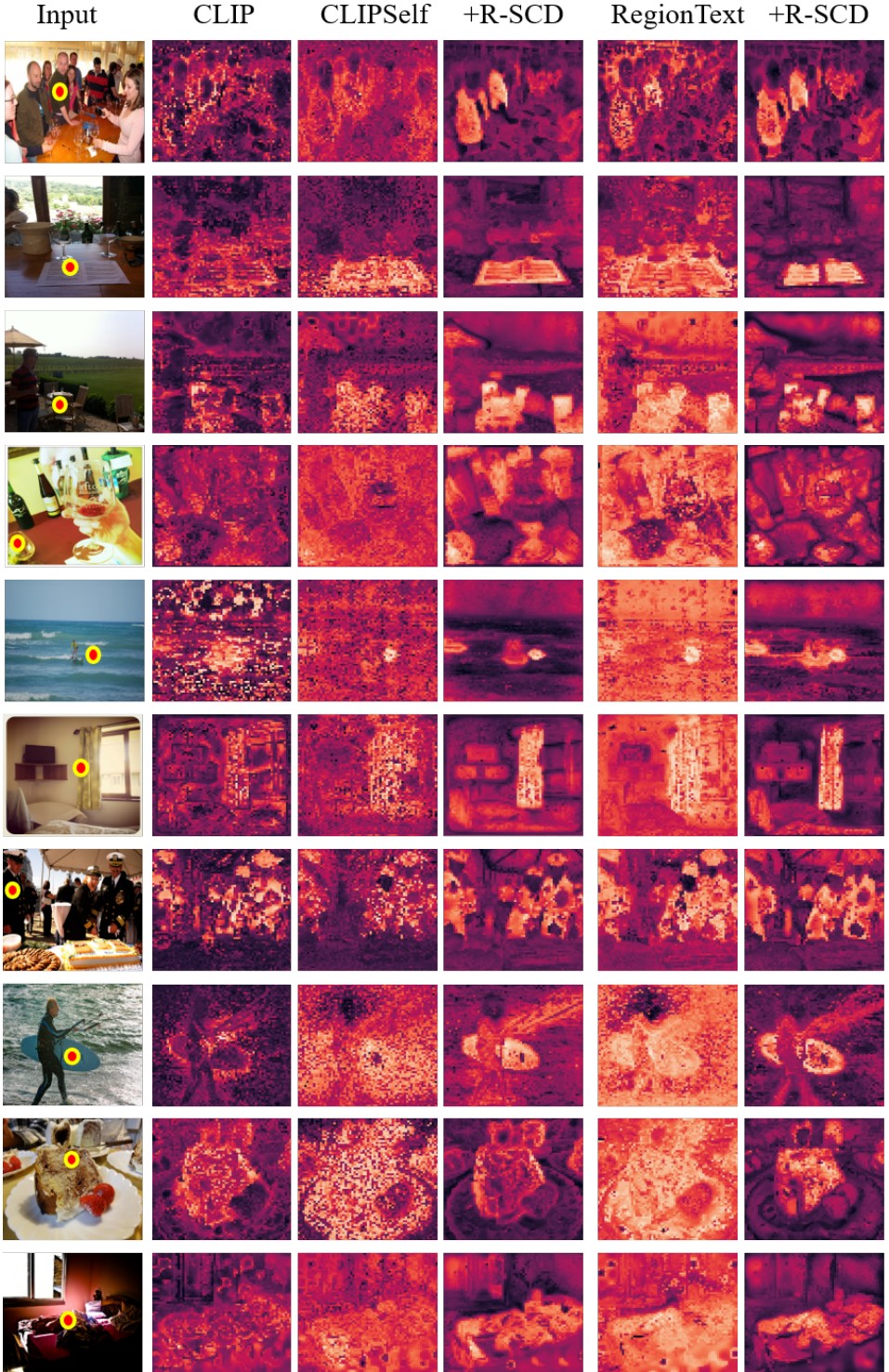

Figure 16: **Visualization of affinity map.** We present the affinity map obtained with the vanilla CLIP, CLIPSelf, R-SC-CLIPSelf, RegionText, and R-SC-RegionText respectively. The query token is marked with a red dot.

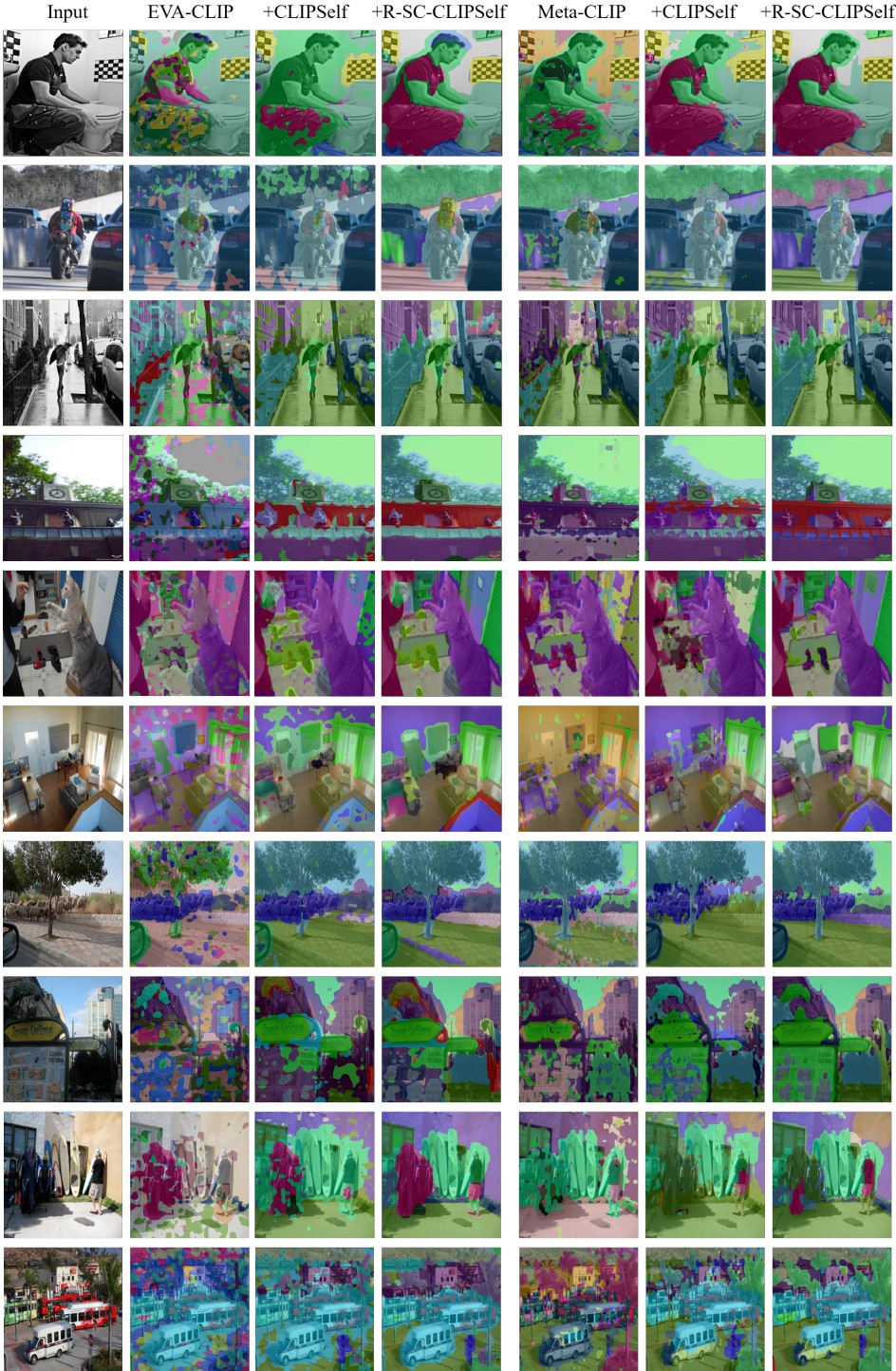

Figure 17: **Visualization of segmentation results with MaskCLIP.** We present the visualization results of MaskCLIP segmentation with EVA-CLIP and Meta-CLIP. Best viewed with color and zoomed in.

