# OpenReview forum: "Refining CLIP's Spatial Awareness: A Visual-Centric Perspective"
_ICLR.cc/2025/Conference — ICLR 2025 Poster_

### Official Review · Reviewer_Mtwu · 2024-10-31

**Soundness:** 3
**Presentation:** 2
**Contribution:** 2
**Rating:** 6
**Confidence:** 4

**Summary:**

This paper aims to enhance CLIP's spatial awareness. That is to say, increases the quality of dense features extracted by CLIP. It proposes the Spatial Correlation Distillation (SCD) framework, which preserves CLIP's inherent spatial structure and mitigates degradation for spatial awareness by Region-Languaeg Alignment. It also introduces a lightweight Refiner that extracts refined correlations directly from CLIP before feeding them into SCD, based on an intriguing finding that CLIP naturally captures high-quality dense features.

**Strengths:**

1. This pager reveals the potential problems of previous works, that "the RLA process projects dense visual embeddings into a text-oriented domain, making them incompatible with visual-centric objectives". To tackle this, the paper proposes to conduct Spatial-Correlation-guided Region-Language Alignment with Refiner to preserve spatial awareness. This design is novel and reasonable.

2. The performance improvement is significant compared with baseline methods.

3. The experiments and analysis are comprehensive.

**Weaknesses:**

1. The paper writing is not so rigorous. Authors should give clear definitions of each term they are discussing. Like what is the definition of "spatial awareness", what it means by a better dense feature, what is "visual-centric", what is "intra-feature structural relationships", etc.

As an example, spatial awareness here is (probably) defined as performance for tasks like localization and recognition, which I think, is equivalent to increasing the quality of dense features extracted by CLIP, which means different parts in the image with different semantics should be extracted with the features that are distinguishable from each other.

**Questions:**

1. It might be better to use "language" or "text" instead of "linguistic"

2. How come methods specifically designed for CLIP dense prediction like CLIPSelf and RegionText work even worse than vanilla CLIP?

---

> ### Author Response · Authors · 2024-11-19
> **Response to Reviewer Mtwu**
>
> We appreciate your valuable comments and questions. We hope that our response can address your concerns.
>
> ***Response to W1. & Q1 (presentation)***: Thanks for your helpful advice, we've strengthen the presentation quality of our paper in the revised version, please kindly refer to the highlighted contents.
>
> ---
>
> ***Response to Q2 (why CLIPSelf and RegionText work worse than vanilla CLIP)***: Thank you for the insightful question. Multi-modal dense prediction tasks demand a dual capability: aligning dense representations with language and preserving visual-centric spatial awareness. RLA methods tend to prioritize the former, often at the cost of the latter. Given CLIP's inherently limited dense-level vision-language alignment, this trade-off can yield relatively good performance in multi-modal tasks.
>
> However, from a visual-centric perspective, relying on language supervision poses significant challenges. The modality gap and the inherent limitations of language supervision often lead to the loss of critical local visual semantics, which is essential for traditional dense prediction tasks that rely on precise localization and recognition. As a result, these methods often perform worse than vanilla CLIP.
>
> In this paper, we argue that spatial awareness is just as crucial as vision-language alignment—a perspective often overlooked in recent RLA literature.  Importantly, we demonstrate that achieving robust region-language alignment does not require compromising a model's spatial awareness. By addressing this balance, our experimental results show improvements in both visual-centric and multi-modal prediction tasks. We hope these findings will inspire future designs of RLA methods and vision-language models.

---

### Official Review · Reviewer_SSGt · 2024-11-02

**Soundness:** 3
**Presentation:** 3
**Contribution:** 3
**Rating:** 6
**Confidence:** 5

**Summary:**

This paper proposes a framework called Spatial Correlation Distillation(SCD) to refine the spatial awareness of CLIP. To recall the visual perception of vanilla CLIP loss in Region-Language Alignment (RLA) training, it proposes a Spatial Correlation Distillation training process and a light-weight refiner to distill the visual relationship from the vanilla CLIP. SCD can improve the existing OV methods and achieve the new sota on the ov tasks.

**Strengths:**

1. Experiments. The experiments are sufficent and solied to prove the improvement of framework.

2. Significance. The framework is a convience, plug and play, and effecitive pipeline to improve the existing methods of the OV tasks.

3. Motivation. The motivation is clear. The Fig.1 and intro explain the loss of dense prediction ability of RLA process.

**Weaknesses:**

1. Novelty. The frame work is composed with SCD and a refiner and the novelty of the SCD is limited. The SCD is a common distillation module to reserve the similarity between the RoI features of student and teacher models.

2. Lack of analysis. This paper does not provide a deeper analysis of the experimental results. For example, why R-SC can effectively improve the classification accuracy in Tab.1? In terms of motivation and structure, this module improves spatial perception and does not have much gain in classified task with the annotated masks..

3. Typo. What is the Fig.2.o in the line 193?

**Questions:**

1. In the SC-RLA, how to ensure the correspondence between the RoI regions from teacher and student models? The student model would be trained, so the number, order and attribute of the RoI regions would become different with the frozen image encoder.

2. In the line 242 -258, would the contents of embeded sampled images {X_i} affect the SPATIAL AWARENESS? Could you provide some quantitative analysis like the cosine similarity between the {X_i} and X_t ?

3. Since the motivation and module is to improve spatial awareness (which can also be seen from the visualization), are there more segmentation related visualizations? Qualitative results using segmentation would be more convincing (e.g. finer edges)

---

> ### Author Response · Authors · 2024-11-19
> **Response to Reviewer SSGt (part 1)**
>
> We appreciate your valuable comments and questions. We hope that our response can address your concerns.
>
> ***Response to W1 (novelty of SCD):*** Our core contributions regarding SCD do not primarily lie in proposing a novel distillation mechanism. More importantly, we identify the necessity of integrating a visual-centric constraint with RLA to mitigate the neglected dense perception degradation issue widely encountered. As discussed in the Related Work section, the potential of correlation distillation in multi-modal scenarios remains largely untapped. Moreover, we also conduct an ablation study in Sec 4.5 to showcase the effects of our SCD’s technical modifications to make correlation distillation compatible with RLAs.
>
> ------
>
> ***Response to W2 & Q2 (more analysis)***: We would like to answer W2 and Q2 together. Our claim is that multi-modal requires both vision-language alignment and robust spatial awareness. While RLAs primarily focus on the former, our approach emphasizes the latter. To support this, we have conducted several analyses highlighting the success of our designs from a visual-centric perspective. These include dense-level t-SNE visualizations (Fig.1), affinity visualizations (Fig.5), and unsupervised segmentation (Fig.5). Our analysis demonstrates that R-SCD significantly enhances the model's dense-level separability and localization capability (The activation highly matches the object contour). Therefore, when integrated with the RLAs, our method exhibits significant improvement including the dense classification task in Tab.1.
>
> Also, according to your suggestion, we hope an additional analysis can further solve your concern (presented in Appendix C.3):
>
> The Refiner is designed to mitigate the semantic contamination from irrelevant context, therefore enhancing the model's sensitiveness to local semantics. To validate this, we concatenate two independently sampled images $X_A$ and $X_B$ side by side, denoted as $X_{AB}$, which introduces context disturbance from $X_B$ to $X_A$. We forward $X_A$  to the image encoder to obtain regional feature map $Z_{A|AB}, Z_{B|AB}, Z_A, Z_B$. We derive the coupling ratio as:
>
> $$ \text{CR} = E_i[\frac{cos(Z_{A|AB}[i], Z_{B|AB}[j])}{cos(Z_{A}[i], Z_{B}[j])}], j = \text{arg}\max_k cos(Z_{A|AB}[i], Z_{B|AB}[k]), $$
>
> where we identify the most similar token $j$ in $Z_{B|AB}$ to the token $i$ in $Z_{A|AB}$, and analyze whether this similarity arises from the coupling of irrelevant semantics introduced by the concatenation operation. Ideally, the CR value is expected to be close to 1, as $X_A$ and $X_B$ possess independent semantics. The measured CR values are reported as below:
>
> | Method           | CR $\downarrow$ |
> | ---------------- | --------------- |
> | EVA-CLIP         | 2.32            |
> | + CLIPSelf       | 1.86            |
> | EVA-CLIP-Refiner | 0.95            |
> | +R-SC-CLIPSelf   | 0.97            |
>
> The results indicate that both the original and CLIPSelf-finetuned CLIP models are significantly affected by semantic coupling. In contrast, our proposed Refiner effectively addresses this issue, demonstrating high consistency with its intended design goals of eliminating semantic contamination.
>
> Similarly, for your question:
>
> > would the contents of embeded sampled images {X_i} affect the SPATIAL AWARENESS ?
>
> Randomly introduced irrelevant background $X_i$ can degrade the spatial awareness of the foreground $X_t$ if semantic contamination is present. To evaluate this effect, we compare the unsupervised segmentation performance of the isolated $X_t$ and the concatenated image $X_t |X_i$ on the Cityscapes dataset, yielding the following results:
>
> | Method /Feature for inference | MIOU                      |
> | ----------------------------- | ------------------------- |
> | EVA-CLIP/ $Z_{X_t}$           | 22.6                      |
> | EVA-CLIP/ $Z_{X_t\|X_i}$      | 22.2 $_{\downarrow 0.4}$  |
> | CLIPSelf/ $Z_{X_t}$           | 17.1                      |
> | CLIPSelf/  $Z_{X_t\|X_i}$     | 16.6  $_{\downarrow 0.5}$ |
> | Refiner/ $Z_{X_t}$            | 25.2                      |
> | Refiner/  $Z_{X_t\|X_i}$      | 25.2                      |
>
> Overall, the above analysis highlights another key aspect of our model's effectiveness: the Refiner mitigates the dominance of dense representations by unintended contextual influences, making them more sensitive to local semantics, more distinguishable, and better suited for dense-level tasks. As a result, the R-SCD pipeline enhances CLIP's image encoder, making it more effective for dense prediction tasks in both open-vocabulary and visual-centric settings.

---

> ### Author Response · Authors · 2024-11-19
> **Response to Reviewer SSGt (part 2)**
>
> ***Response to W3***: Sorry for the typo, which should be “Fig.2. To …”. We’ve corrected this line in the revised version.
>
> ------
>
> ***Response to Q1 (consistency of RoI)***: There are two settings for defining region proposals: (a) **Random Proposals**: Proposals are randomly sampled and consistently applied to both the student and teacher encoders; (b) **RPN Proposals**: Following CLIPSelf, the RoI regions are pre-generated using an additional RPN structure before fine-tuning. These approaches ensure that misalignment of the RoIs is not a concern.
>
> ---
>
> ***Response to Q3 (segmentation related visualizations)***: Thank you for your valuable suggestion. In response, we have incorporated the off-the-shelf MaskCLIP [1] as the evaluation protocol and included visualized segmentation results in Fig. 7 and Fig. 17 of the revised manuscript. We believe this addition better demonstrates the generalizability of our approach and provides clearer clarification.
>
> **Reference**
>
> [1] Zhou et al. "Extract free dense labels from clip."  ECCV 2022.

---

> > ### Comment · Reviewer_SSGt · 2024-11-29
> >
> > Thanks for the detailed feedback. My concern has been addressed.

---

> > > ### Author Response · Authors · 2024-11-29
> > >
> > > Thank you very much for your response. We greatly appreciate your valuable insights, and we are pleased that our clarifications have effectively addressed your concerns. We will continue to engage in the rebuttal process. Should you have any additional questions or suggestions, please do not hesitate to share them. We remain open to any further feedback.

---

### Official Review · Reviewer_FR8o · 2024-11-04

**Soundness:** 3
**Presentation:** 3
**Contribution:** 3
**Rating:** 6
**Confidence:** 4

**Summary:**

This paper aims to improve upon the Region-Language Alignment (RLA) approaches for Contrastive Language-Image Pre-training (CLIP) models. In order to not only promote the linguistic alignment but also preserve the spatial awareness, Spatial Correlation Distillation (SCD) is proposed to plug into the existing methods such as RegionCLIP and CLIPSelf. Refiner is also introduced to enhance the regional representation of teacher model. The experiments on the open-vocabulary dense prediction tasks demonstrate the effectiveness of the proposed method.

**Strengths:**

Motivated by the observation that RLA methods suffer from notable loss in spatial awareness for CLIP ViTs, SCD is specifically designed to capture the spatial structure in a region. The widespread experiments show the superior performance of the proposed method across multiple open-vocabulary dense prediction benchmarks.

**Weaknesses:**

The motivation to design Refiner has been implied in CLIPSelf. When K = N, the proposed Refiner is almost the same as CLIPSelf, which reduces the technical novelty. Also, the ablation study of K is absent in this work.

The application of the proposed method is restricted. Spatial Correlation Distillation (SCD) is auxiliary and is to preserve the spatial awareness when another optimization is applied (e.g. RLA). Therefore, it seems that SCD cannot be applied independently since in this case the weights of teacher model and student model are always equal. Besides, R-SC-V is only learned from the teacher model that is optimized by Refiner (similar to CLIPSelf), so it cannot be further applied to CLIPSelf based approach.

In order to showcase the generalizability and scalability of the proposed method, the experiments with data scaling up are expected to be provided, which is missing in the current version.

**Questions:**

Please see weaknesses.

---

> ### Author Response · Authors · 2024-11-19
> **Response to Reviewer FR8o (part 1)**
>
> We appreciate your valuable comments and questions. We hope that our response can address your concerns.
>
> ***Response to W1***: (**Refiner vs CLIPSelf)**
>
> We appreciate your feedback and would like to clarify that even when $K=N$, $i.e.$ finetuning the entire image encoder as the Refiner, there are fundamental distinctions between the two training strategies:
>
> ***(a)  Motivation***:
>
> CLIPSelf: CLIPSelf observed that region patches lack consistent alignment with text representations. Their motivation is to address this gap by using the CLS token, which encapsulates global semantic information, to guide and refine region-level features, thereby improving region-level text alignment.
>
> Ours:  We highlight a different issue—CLIP's lack of spatial awareness, particularly the localization and recognition degradation of dense representations in dense tasks due to its focus on image-level alignment with text. This inherent noise in CLIP ViTs' spatial representation is identified as a core limitation.
>
> ***(b) Strategy:***
>
> To provide a clearer comparison, we recall the respective objectives:
>
> $$ L_{CLIPSelf} = L_{Align}(\text{RoIPool}(Z_i), CLS_i), \\ L_{Refiner} = \sum_j L_{Align}(Z_i[j], Z'_i[j]), $$
>
> where $Z_i$ is the i-th regional feature map.
>
> CLIPSelf: It extracts **region** representations using RoI pooling and aligns them with the $[CLS]$ token as supervision. While this approach appears similar to our "global-to-local" dynamic, the supervision signal in CLIPSelf comes from the $[CLS]$ token, which is already aligned to the text domain. This alignment inherently discards **crucial local spatial details**, leading to significant visual-centric degradation—a core limitation of CLIPSelf that we aim to address.
>
> Ours: In contrast, our method leverages **point-level** representations $Z[j]$ with finer granularity that better encodes visual-centric knowledge. By adopting the corresponding feature $Z'[j]$ from a local crop as the teacher, we enable a purely **visual-centric** fine-tuning strategy, preserving high-quality local semantics.
>
>  ***(c) Experimental Results***:
>
> As mentioned in the motivation, CLIPSelf only targets the alignment between text and regions, ignoring the spatial correlation among the patches.  To substantiate this difference, we conduct an additional ablation where the Refiner is fine-tuned using CLIPSelf's objective. As shown below, the output quality of the Refiner-CLIPSelf significantly declines under the CLIPSelf-based objective. Additionally, our qualitative analysis (see Fig. 16; Fig. 15 in the original version) highlights that the local representations from the CLIPSelf-based model share high similarity with surrounding patches, even those without relevant semantics, being dominated by global semantics. This behavior demonstrates its inability to serve as effective visual-centric local supervision. We have also included further empirical studies in the revised Appendix C.3.
>
> | Method          | CLIP | Refiner-CLIPSelf | Refiner-Ours |
> | --------------- | ---- | ---------------- | ------------ |
> | Citiscapes mIoU | 22.6 | 11.5             | 25.2         |
>
> >  Ablation on K
>
> In our original version, we conducted an ablation study on the depth of the Refiner ($K$), but deferred it to Appendix E (Tab. 12 in the revised version) due to space constraints. For your convenience, we provide the details below:
>
> We investigate the impact of the depth of the Refiner on the performance of the distilled model. The depth will affect the distillation process from two aspects: i) the balance between the capacity of refining and preserving the original visual knowledge learned by the visual encoder, and ii) the computational efficiency of the training process. We conduct experiments with different depths of the Refiner. A deeper Refiner will increase the parameter size and the complexity of the fine-tuned model, but more difficult to preserve learned knowledge of the pre-trained model. The model with a 4-layer Refiner achieves the best performance, obtaining balance between the refining capacity and knowledge preservation.
>
> | K Layers | 2    | 3    | 4    | 5    |
> | -------- | ---- | ---- | ---- | ---- |
> | OV-COCO  | 40.3 | 40.7 | 40.9 | 40.5 |
>
> ------

---

> ### Author Response · Authors · 2024-11-19
> **Response to Reviewer FR8o (part 2)**
>
> ***Response to W2 (applicability)***: We appreciate the reviewer’s insights and would like to address the concerns regarding R-SCD’s applicability.
>
> > The application of the SCD is restricted.
>
> RLAs are important since the dense-level potential of VLMs has not been fully explored, especially in the era of LLM.  SCD serves as a valuable visual-centric complement to RLAs, especially when integrated with the Refiner, offering a pathway to designing novel VLMs with enhanced spatial awareness. We believe this integration has the potential to enable a wide range of applications in future research. For instance, large VLMs such as Qwen-VL [1] and BLIP-2 [2] rely on the CLIP image encoder to capture fine-grained visual information for their LLM modules. In this context, R-SCD-based RLAs could serve as a potentially feasible method to address CLIP’s dense-level limitations, thereby further improving the performance of these advanced VLMs.
>
> Regarding the concern about "equal weights of teacher and student models," this is not a necessary condition for R-SCD to function. Since R-SCD is a **relational constraint**, it does not require the student and teacher to share the same embedding space, parameters, or architecture. For instance, we demonstrate that using DINO V2 + Refiner to enhance CLIPSelf-based EVA-CLIP results in effective improvements:
>
> | Method                | OV-COCO |
> | --------------------- | ------- |
> | CLIPSelf              | 37.6    |
> | R-SC-CLIPSelf         | 40.9    |
> | R-SC-CLIPSelf-DINO V2 | 41.6    |
>
> > Application of R-SC-V.
>
> R-SC-V is specifically designed to enhance **image-only dense prediction tasks** (e.g., segmentation and detection) and is not intended for CLIPSelf-based methods. For example, applying R-SC-V to DINO V2 demonstrates its effectiveness. To further explore its potential, we integrated R-SC-V with MAE for unsupervised segmentation, yielding the following results:
>
> | Method  | mIoU | pACC |
> | ------- | ---- | ---- |
> | MAE     | 21.5 | 59.1 |
> | +R-SC-V | 24.6 | 63.0 |
>
> These results highlight R-SC-V’s ability to enhance spatially-aware dense embeddings and its potential benefits to the SSL community. We will include additional results and insights in future work to further expand the understanding and applicability of our method.
>
> ------
>
> ***Response to W3: (generalizability and scalability)*** We additionally conduct two experiments to address your concerns: (a) for **generalizability**, we perform R-SCD on more VLMs like DFN [3] and Meta-CLIP [4]. (b) for **scalability**, following CLIPSelf,  we train R-SC-CLIPSelf on CC3M dataset for 1 epoch. We've added the results in our revised version. We adopt the off-the-shelf zero-shot segmentation as in MaskCLIP [5] as it's training-free and directly reflects both spatial awareness and vision-language alignment quality.
>
> (a) generalizability: training on more VLMs:
>
> | VLM Model      | PASCAL Context | COCO Stuff |
> | -------------- | -------------- | ---------- |
> | OpenAI-CLIP    | 25.5           | 14.6       |
> | +CLIPSelf      | 26.4           | 16.1       |
> | +R-SC-CLIPSelf | 27.9           | 17.5       |
> | DFN            | 29.4           | 18.6       |
> | +CLIPSelf      | 30.8           | 20.1       |
> | +R-SC-CLIPSelf | 32.1           | 21.2       |
> | Meta-CLIP      | 30.3           | 20.0       |
> | +CLIPSelf      | 30.1           | 19.7       |
> | +R-SC-CLIPSelf | 33.6           | 22.0       |
> | EVA-CLIP       | 22.8           | 15.6       |
> | +CLIPSelf      | 32.2           | 20.1       |
> | +R-SC-CLIPSelf | 37.0           | 23.8       |
>
> (b) scalability: training on CC3M:
>
> | Method/ Dataset      | PASCAL Context | COCO Stuff |
> | -------------------- | -------------- | ---------- |
> | R-SC-CLIPSelf / COCO | 37.0           | 23.8       |
> | R-SC-CLIPSelf / CC3M | 38.2           | 25.0       |
>
>  **Reference**
>
> [1] Bai et al. "Qwen-vl: A frontier large vision-language model with versatile abilities." ArXiv 2023.
>
> [2] Li et al. "BLIP-2: Bootstrapping Language-Image Pre-training with Frozen Image Encoders and Large Language Models." ICML 2023.
>
> [3] Fang et al. "Data filtering networks." ICLR 2024.
>
> [4] Xu et al. "Demystifying clip data." ICLR 2024.
>
> [5] Zhou et al. "Extract free dense labels from clip."  ECCV 2022.

---

### Author Response · Authors · 2024-11-19
**Global Response**

We sincerely thank all the reviewers for their thoughtful and constructive feedback, which has been both encouraging and insightful. We are pleased to see that the reviewers appreciate the following aspects of our work:

- The effectiveness of our designs, demonstrating significant performance improvements (FR8o, SSGt, Mtwu)
- The solid and comprehensive nature of our experiments (SSGt, Mtwu)
- The novelty and soundness of our design, supported by thorough analysis (Mtwu)
- The framework’s convenience and plug-and-play capability (SSGt)

We have carefully considered the reviewers' comments and revised our manuscript accordingly. The updated content is highlighted in **magenta** for clarity. Below is a brief outline of the revisions:

- In response to Mtwu, we have provided clearer explanations of the crucial concepts in our paper
- Following MaskCLIP [1], we have added a new off-the-shelf zero-shot segmentation experiment in Sec. 4.3, including additional VLMs such as Meta-CLIP [2] and DFN [3], to demonstrate the generalizability of our method. We have also included corresponding visualizations following SSGt’s suggestion.
- At SSGt's suggestion, we have conducted further empirical analysis in Appendix C.3 to better interpret the positive effects of our model.

**Reference**

[1] Zhou et al. "Extract free dense labels from clip."  ECCV 2022.

[2] Fang et al. "Data filtering networks." ICLR 2024.

[3] Xu et al. "Demystifying clip data." ICLR 2024.

---

### Meta-Review · Area_Chair_fmk2 · 2024-12-16

**Metareview:**

The paper presents the Spatial Correlation Distillation (SCD) framework, aimed at enhancing CLIP’s spatial awareness to overcome its limitations in dense prediction tasks. The authors propose the addition of a lightweight Refiner module, which extracts and refines spatial correlations from CLIP’s inherent dense features, boosting performance in tasks requiring spatial precision, such as segmentation and object detection.

AC concurred with the reviewers on the significant contribution of improving CLIP’s spatial awareness and commended the authors’ efforts in addressing concerns, particularly in clarifying the technical methodology and enhancing the quality of the presentation. Reviewer Mtwu highlighted the lack of clear definitions for key terms in the paper and suggested that the authors provide more detailed explanations to further improve readability. AC encourages the authors to release the code to support the community and enhance reproducibility.

**Additional Comments On Reviewer Discussion:**

The initial reviewers' concerns focus on (1)The effectiveness of designs, demonstrating significant performance improvements (FR8o, SSGt, Mtwu) (2)The solid and comprehensive nature of our experiments (SSGt, Mtwu) (3)The novelty and soundness of our design, supported by thorough analysis (Mtwu) (4)The framework’s convenience and plug-and-play capability (SSGt). The authors have actively addressed these concerns, resolving the majority of them effectively.

---

### Decision · Program_Chairs · 2025-01-22

Accept (Poster)